# A network of interacting ciliary tip proteins with opposing activities imparts slow and processive microtubule growth

Harriet A. J. Saunders [1,6], Cyntha M. van den Berg[1,6], Robin A. Hoogebeen[1], Donna Schweizer [1], Kelly E. Stecker[2,3], Ronald Roepman [4], Stuart C. Howes [5] & Anna Akhmanova [1]✉

Cilia are motile or sensory organelles present on many eukaryotic cells. Their formation and function rely on axonemal microtubules, which exhibit very slow dynamics, but the underlying mechanisms are largely unexplored. Here we reconstituted in vitro the individual and collective activities of the ciliary tip module proteins CEP104, CSPP1, TOGARAM1, ARMC9 and CCDC66, which interact with each other and with microtubules and, when mutated in humans, cause ciliopathies such as Joubert syndrome. We show that CEP104, a protein with a tubulin-binding TOG domain, and its luminal partner CSPP1 inhibit microtubule growth and shortening. Another TOG-domain protein, TOGARAM1, overcomes growth inhibition imposed by CEP104 and CSPP1. CCDC66 and ARMC9 do not affect microtubule dynamics but act as scaffolds for their partners. Cryo-electron tomography demonstrated that, together, ciliary tip module members form plus-end-specific cork-like structures that reduce protofilament flaring. The combined effect of these proteins is very slow processive microtubule elongation, which recapitulates axonemal dynamics in cells.

Cilia are motile or sensory organelles present on the surface of many eukaryotic cells and their disfunction is associated with numerous diseases collectively called ciliopathies[1–4]. The core of each cilium is formed by a microtubule (MT)-based structure, the axoneme[5]. Diverse proteins responsible for axoneme formation have been identified, yet the biochemical mechanisms governing the dynamics of axonemal MTs remain very poorly understood[6].

Unlike cytoplasmic MTs, axonemal plus ends grow very slowly, likely impeding the formation of a stabilizing guanosine triphosphate (GTP) cap, which is required for preventing MT disassembly[7] and efficient tubulin addition[8]. The absence of a stable GTP cap can be compensated for by specific MT-binding proteins. Excellent candidates for this function in cilia are the components of the ciliary tip interaction network or 'module', CEP104, CSPP1, TOGARAM1, ARMC9 and CCDC66 (ref. 9). Mutations in the genes encoding most of these proteins lead to Joubert syndrome, characterized by brain malformations, breathing problems and other defects[3,9–14]. According to cell biological, biochemical and yeast two-hybrid assays, the ciliary tip module (CTM) proteins interact with MTs and with each other and are all important for controlling ciliary length and the signaling pathways dependent on primary cilia[9,10,15–21]. Some CTM proteins also associate with centrosomes, centriolar satellites and

[1]Cell Biology, Neurobiology and Biophysics, Department of Biology, Faculty of Science, Utrecht University, Utrecht, The Netherlands. [2]Biomolecular Mass Spectrometry and Proteomics, Bijvoet Center for Biomolecular Research and Utrecht Institute for Pharmaceutical Sciences, Utrecht University, Utrecht, The Netherlands. [3]Netherlands Proteomics Center, Utrecht, The Netherlands. [4]Department of Human Genetics, Research Institute for Medical Innovation, Radboud University Medical Center, Nijmegen, The Netherlands. [5]Structural Biochemistry, Department of Chemistry, Bijvoet Centre for Biomolecular Research, Utrecht University, Utrecht, The Netherlands. [6]These authors contributed equally: Harriet A. J. Saunders, Cyntha M. van den Berg. ✉e-mail: a.akhmanova@uu.nl

cytoplasmic MTs and participate in cilia-independent processes such as cell division[17,22–25].

CEP104 (also known as FAP256), TOGARAM1 (also known as Crescerin or CHE12) and ARMC9 are conserved across eukaryotes, where they participate in the biogenesis of motile or sensory cilia[19,26–29]. Both CEP104 and TOGARAM1 have evolutionary conserved TOG domains (Fig. 1a). TOGARAM1 contains four TOG domains, two of which can promote tubulin polymerization in vitro in light-scattering experiments[19], and the same is true for the single tubulin-binding TOG domain of CEP104 (refs. 30–32). Loss-of-function mutations in the genes encoding TOGARAM1 and CEP104 result in short cilia in multiple organisms ranging from flagellated green algae to mammals[9,15,19,26–29,32]. Similarly, loss of CSPP1, a protein that in vitro increases MT stability by binding inside the MT lumen and promoting pausing[33,34], results in short cilia in vertebrates[15]. ARMC9 on its own cannot bind to MTs in cells but acts as a scaffold for other CTM components[9]. ARMC9 and TOGARAM1 have been localized to motile cilia of *Chlamydomonas* and *Tetrahymena*, where they were found to have opposite effects on MT length[27,29], although mutants of both proteins make mammalian primary cilia shorter[9]. Lastly, according to immunoprecipitation assays, CSPP1 and CEP104 interact with CCDC66 and codepletion of CCDC66 with either CSPP1 or CEP104 in cultured human cells results in a further reduction in ciliary length compared to the removal of either CSPP1 or CEP104 alone[20].

Here, we set out to uncover the biochemical mechanisms underlying the function of CTM proteins. We reconstituted in vitro their individual and collective effects on MT dynamics. We found that CEP104 specifically inhibited MT plus-end elongation and shortening and its activity was potentiated by EB3, CSPP1 and CCDC66, which could all recruit CEP104 to MTs. TOGARAM1 functioned as a polymerase that converted the inhibition of MT polymerization, imposed by CEP104, to slow growth. ARMC9 and CCDC66 enhanced the effects of their binding partners. The combination of all five CTM proteins resulted in very slow and processive MT plus-end elongation, an effect that required TOGARAM1 and became less robust when one of the other proteins was left out. MT growth rates in these conditions were in the same range as initial elongation rates of regenerating flagella of single-celled organisms[35–37]. Cryo-electron tomography (cryo-ET) showed that, together, CTM proteins formed a champagne cork-like structure that protruded from MT plus ends and diminished protofilament flaring. Altogether, our findings demonstrate that, through a combination of opposing activities, CTM components can stabilize MT plus ends and drive their slow but persistent elongation.

## Results

### CTM proteins have distinct effects on MT dynamics

To characterize the effects of CTM proteins on MT polymerization, we N-terminally tagged them with mCherry or GFP and purified them from transiently transfected HEK293T cells (Fig. 1a and Extended Data Fig. 1a). Size-exclusion chromatography (SEC) detected no protein aggregation

and mass spectrometry (MS)-based analysis demonstrated that the protein preparations contained only very minor contaminations with other known regulators of MT dynamics, such as CSPP1 (Extended Data Fig. 1b,c). In addition, heat-shock protein Hsp70, a protein that, to our knowledge, has no effect on MT dynamics, was present in the preparations of CCDC66 and CSPP1 (Extended Data Fig. 1c). We used these proteins to perform in vitro reconstitution assays with MTs grown from GMPCPP-stabilized seeds and observed their impact on MT dynamics by total internal reflection fluorescence (TIRF) microscopy[34,38]. Although axonemes are composed of MT doublets, single MTs are appropriate substrates to study the effects of CTM proteins because doublets become singlets at ciliary tips[39–41]. To detect MTs, we used either fluorescent tubulin or fluorescently tagged EB3, which increases MT growth rate and induces catastrophes[42] (Fig. 1b–f). Studying the effects of CTM proteins in the presence of EBs is relevant because EB1 and EB3 localize to axonemal MTs in both primary and motile cilia[41,43,44].

Among the tested proteins, ARMC9 was the only one that displayed no MT binding even when present at 300 nM (Fig. 1g,h), in agreement with data in cells[9]. CCDC66, which is known to interact with MTs[23], bound along the MT lattice already at 10 nM and slightly decreased the growth rate but did not affect the frequencies of catastrophes and rescues and did not induce pausing (Fig. 1d–f,i,j).

CEP104 could not autonomously bind MTs at 10 nM in assays with tubulin alone (Extended Data Fig. 1d). However, at 100 nM, it efficiently blocked MT growth specifically at the plus end (Fig. 1d–f,k), as confirmed by including in the assay a constitutively active fragment of the plus-end-directed kinesin 1 DmKHC (1–421) (Extended Data Fig. 1e). In the presence of its binding partner EB3 (ref. 16), CEP104 could specifically inhibit MT plus-end polymerization already at 10 nM (Fig. 1d–f,l). CEP104 colocalized with EB3 at some growing plus and occasionally also minus ends; however, this did not alter growth rates at the plus end (Fig. 1d,l). Once MT growth was arrested, no regrowth was detected for the remainder of the experiment (Fig. 1e,k,l).

Episodes of inhibited growth, albeit transient ones, were also observed with CSPP1, as shown previously (Fig. 1d–f,m,n)[34]. Both with and without EB3, CSPP1 preferentially binds at both plus and minus ends when they undergo growth perturbations[33,34]; therefore, it displayed discrete sites of increased lattice accumulation (Fig. 1m,n). At the plus ends, CSPP1 accumulations decreased growth rate, induced pauses and facilitated transitions from pause to growth, whereas pausing was less obvious at the minus ends because of their overall slower dynamics (Fig. 1d–f,m,n).

Lastly, TOGARAM1 bound to MT plus ends, reduced their growth rate and induced rescues and these effects were not altered by the presence of EB3 (Fig. 1d–f,o,p). A decrease in MT growth rate was unexpected as TOGARAM1 structurally resembles the MT polymerase ch-TOG/XMAP215 (ref. 19), which accelerates MT polymerization[45]. Altogether, we show that all the full-length members of the CTM can be purified and used for in vitro assays, where they display opposing effects on MT dynamics.

**Fig. 1 | CTM proteins have distinct effects on MT dynamics in vitro. a**, Schematic representation of CTM members and summary table highlighting individual protein effects on MT dynamics. MTB, MT-binding domain; PD, pause domain. **b,c**, Fields of view (left; scale bar: 2 μm) and kymographs (right; scale bars: 2 μm and 60 s) illustrating MT dynamics from GMPCPP-stabilized seeds with either 15 μM tubulin supplemented with 3% HiLyte-488-labeled tubulin or 20 nM GFP–EB3. **d–f**, Parameters of MT plus-end dynamics in the presence of 15 μM tubulin alone or with 20 nM EB3 in combination with indicated concentrations of proteins (from kymographs shown in **b,c,i–p** and Extended Data Fig. 1d). **d**, Bars represent pooled data from three independent experiments (growth rate) or averaged means from three independent experiments (pause and block duration); total number of growth events, pauses/blocks: tubulin alone, *n* = 356, 0; 10 nM CCDC66, *n* = 411, 0; 10 nM CEP104, *n* = 306, 0; 100 nM CEP104, *n* = 0, 138; 10 nM CSPP1, *n* = 422, 97; 10 nM TOGARAM1, *n* = 347, 0; EB3 alone, *n* = 938,

0; EB3 with 10 nM CCDC66, *n* = 562, 0; EB3 with 10 nM CEP104, *n* = 213, 101; EB3 with CSPP1, *n* = 1715, 273; EB3 with TOGARAM1, *n* = 861, 0. ****$P$ < 0.0001; NS, not significant (Kruskal–Wallis test followed by Dunn's post hoc test; all conditions compared to their relevant control (either tubulin alone or tubulin with 20 nM EB3)). Videos were acquired for 10 min; therefore, this is the maximum time for pause duration. In **e,f**, bars represent averaged means from three independent experiments. Error bars represent the s.e.m. **g–p**, Fields of view (left; scale bar: 2 μm) and kymographs (right; scale bars: 2 μm and 60 s) illustrating MT dynamics from GMPCPP-stabilized seeds either with 15 μM tubulin supplemented with 3% HiLyte-488-labeled or rhodamine-labeled tubulin or with 20 nM GFP–EB3 or mCherry–EB3 and indicated concentrations and colors of CTM proteins. Orange arrowheads, blocked plus ends; orange arrows, CEP104-tracking minus ends; yellow arrowheads, pauses; blue arrowheads, rescues.

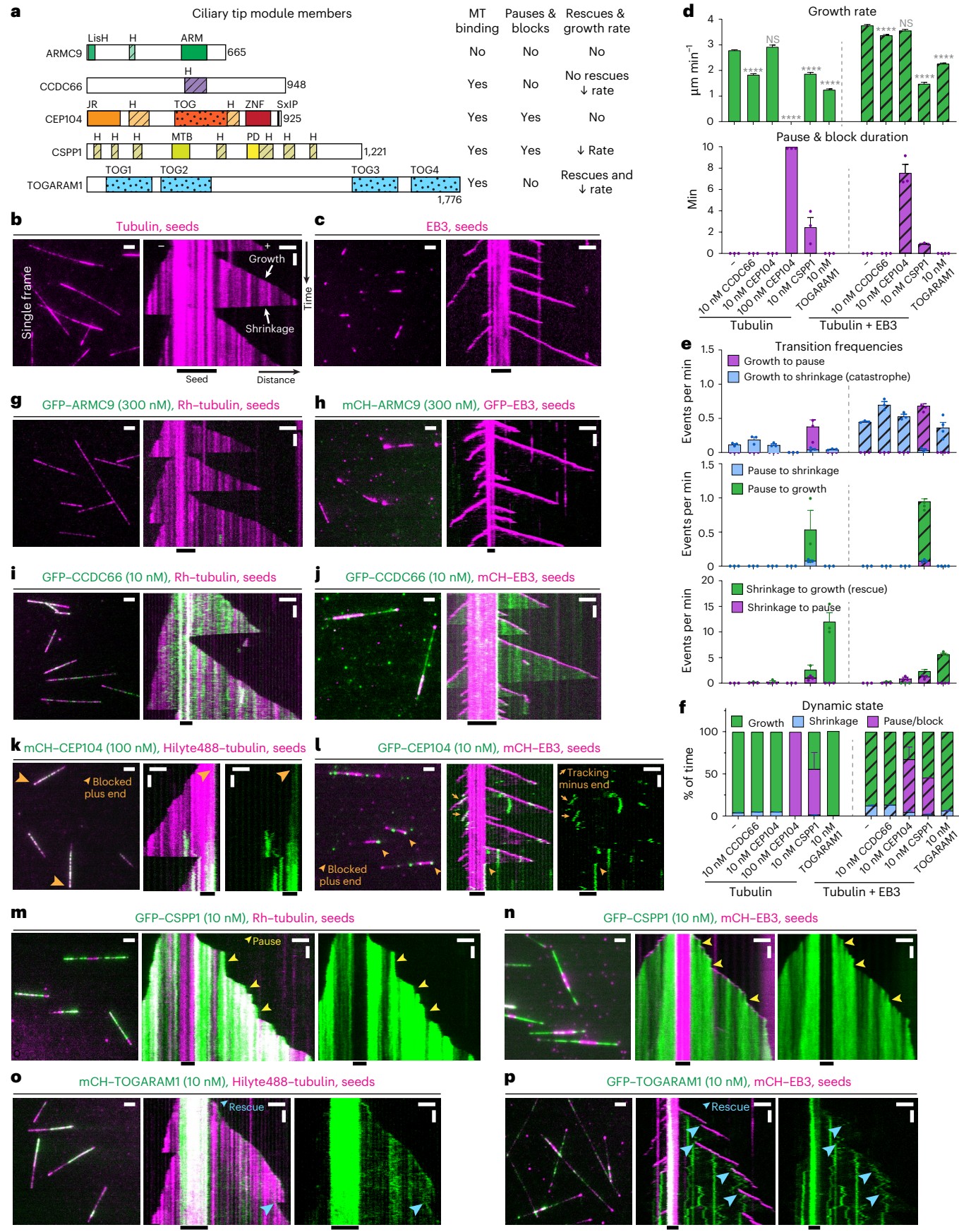

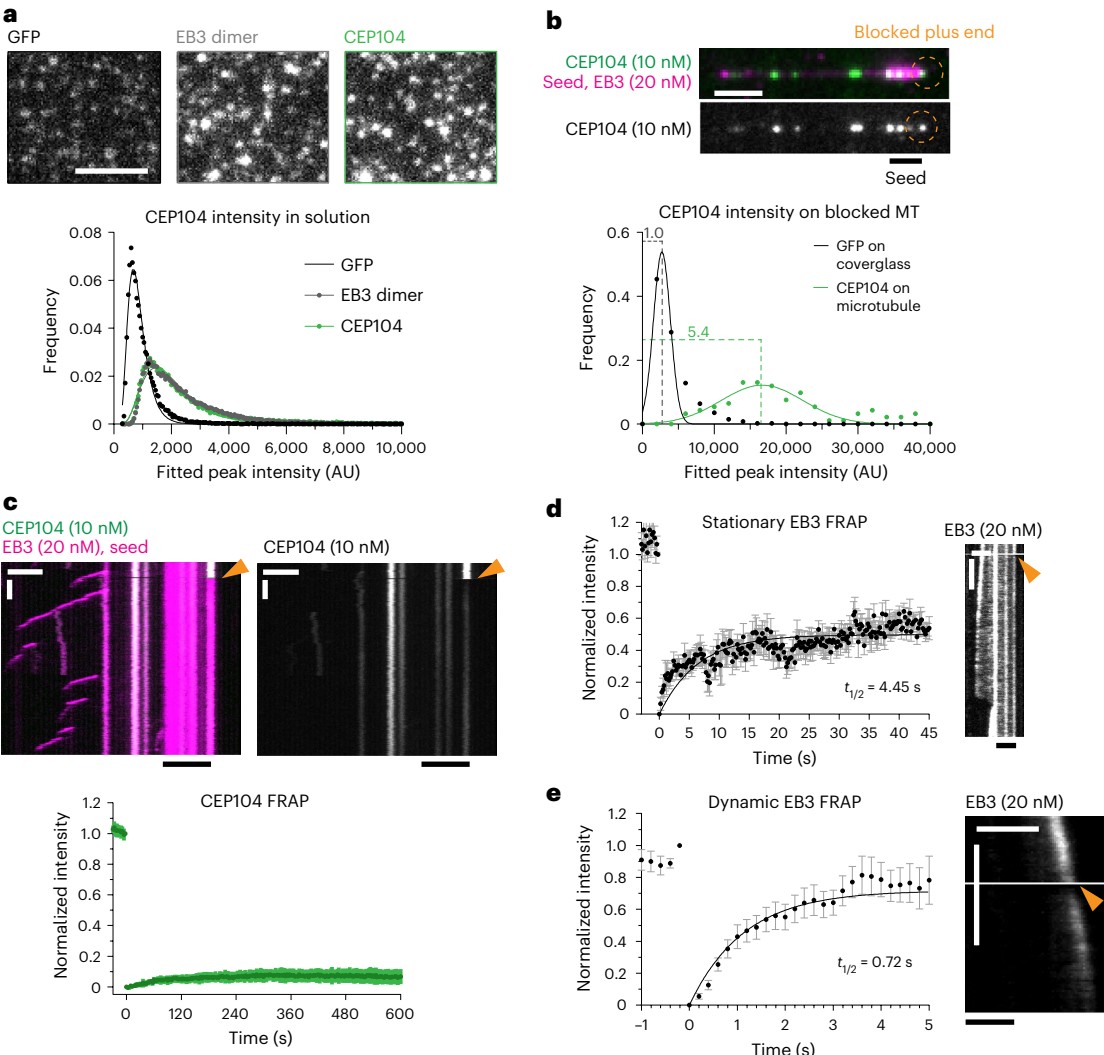

**Fig. 2 | A few CEP104 molecules stably block MT plus ends. a**, Fields of view (top) and histogram plot (bottom) of fluorescence intensities of single GFP molecules, GFP–EB3 dimers and GFP–CEP104 dimers immobilized in separate chambers of the same coverslip. Number of molecules analyzed: GFP, $n = 29{,}981$; GFP–EB3, $n = 55{,}378$; GFP–CEP104, $n = 32{,}335$. Scale bar: 2 µm. **b**, Representative MT with CEP104-blocked plus end (top) and histogram plot (bottom) of fluorescence intensities of single GFP molecules and GFP–CEP104 intensity at blocked plus end immobilized in separate chambers of the same coverslip. Number of molecules analyzed: GFP, $n = 13{,}925$; GFP–CEP104, $n = 92$. Scale bar: 2 µm. **c–e**, FRAP analysis of CEP104 (**c**) and EB3 (**d**) at blocked MT plus ends or dynamic EB3 at growing plus ends (**e**). The arrowhead marks the point of photobleaching in representative kymographs. Scale bars: 2 µm and 60 s (**c**); 2 µm and 10 s (**d**,**e**). Plots show averaged curves with exponential fit. Number of FRAP measurements: CEP104, $n = 17$; stationary EB3, $n = 14$; dynamic EB3, $n = 10$. Error bars represent the s.e.m.

## Characterization of MT plus-end blocking by CEP104

Among the CTM proteins analyzed above, the effect of CEP104 was the most unexpected as it was thought to be an MT polymerase[32]; therefore, we set out to dissect it in more detail. Single-molecule counting experiments demonstrated that CEP104 forms a dimer, in agreement with the sedimentation profile of its first helical (H) domain (Fig. 2a)[30]. In the presence of EB3, ~5–6 CEP104 molecules (2–3 dimers) were sufficient to block MT plus-end growth (Fig. 2b). In these conditions, fluorescence recovery after photobleaching (FRAP) showed that CEP104 did not exchange at blocked MT plus ends (Fig. 2c), whereas EB3, associated with the same ends, did partially turn over, albeit significantly slower than on growing MT plus ends (Fig. 2d,e). This could be explained by the direct interaction of EB3 and CEP104 on the MT through the SxIP motif of CEP104, SKIP (Fig. 3a)[16]. Mutations impacting this motif are known to abolish the interactions between the EBs and their partners[32,46]. Indeed, double substitution of CEP104's SxIP motif, SKIP to SKNN, prevented EB3-dependent recruitment of CEP104 to MTs; therefore, even in the

presence of EB3, 100 nM CEP104 SKNN was needed to block plus-end growth (Fig. 3a–d, Extended Data Fig. 2a).

Furthermore, the single TOG domain of CEP104 was required to inhibit MT growth. CEP104 lacking the TOG domain could still be recruited to MTs by EB3 but no plus-end blocking was observed (Fig. 3a,b,e and Extended Data Fig. 2a). In the presence of EB3, CEP104ΔTOG could bind to growing MT ends and accumulate at the border between the seed and the plus-end-grown lattice (Fig. 3e), a localization for which we currently have no explanation.

Because recent studies described functional and biochemical interactions between CEP104, CCDC66, CSPP1 and ARMC9 (refs. 9,15,20), we next tested in vitro the combinations of these proteins. Here, 2 nM CEP104 on its own, in the presence of EB3 or together with ARMC9 showed hardly any MT binding, whereas, in the presence of 15 nM CCDC66, 2 nM CEP104 was already sufficient to occasionally block MT plus-end outgrowth from the seed (Fig. 3f,g and Extended Data Fig. 2b). CEP104-driven growth inhibition at MT seeds became even more common at 10 nM CEP104 with 15 nM CCDC66 and, in these

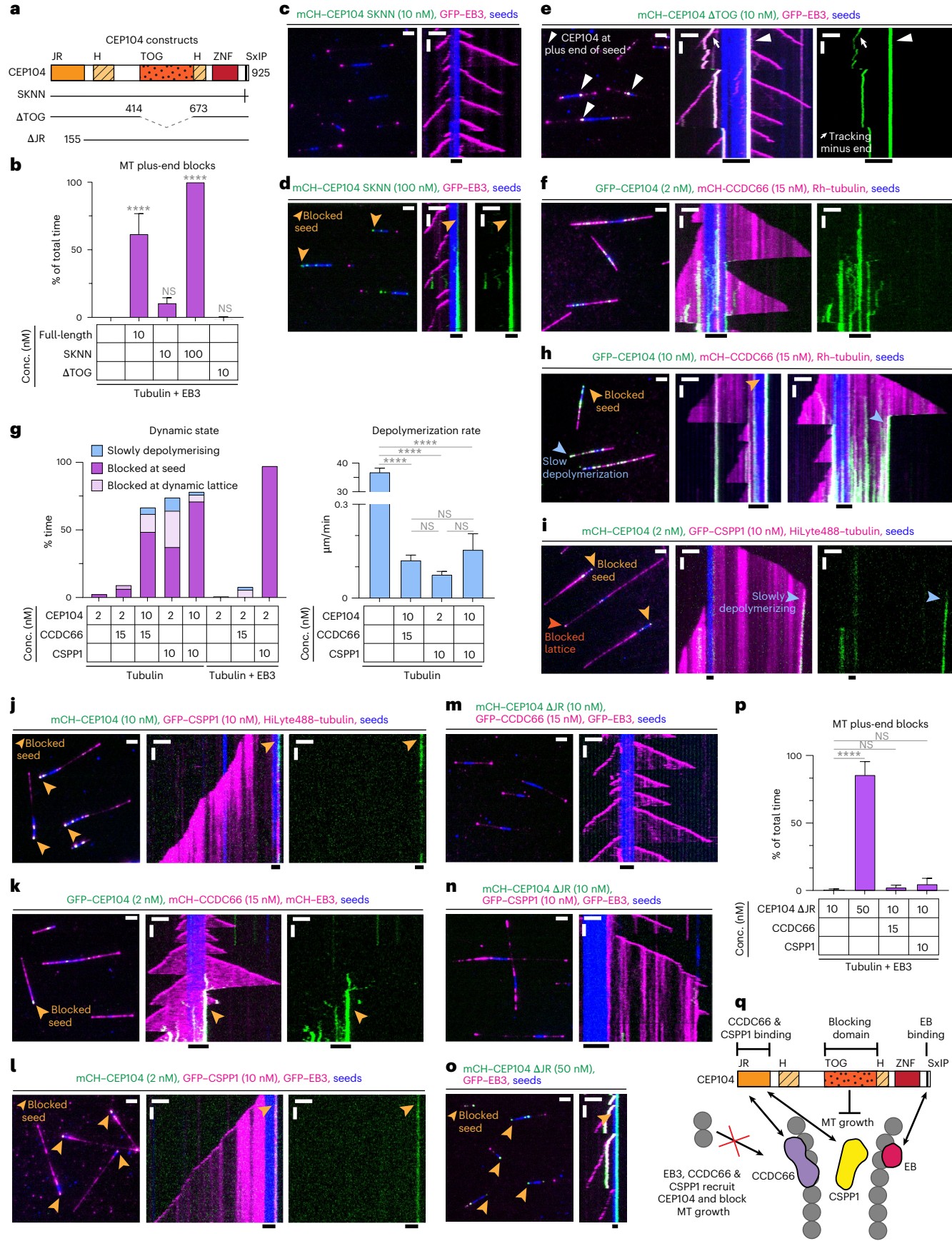

**Fig. 3 | Plus-end blocking by CEP104 is potentiated by EB3, CCDC66 and CSPP1.** **a**, Scheme of CEP104 constructs. **b**, Percentage of time MT plus ends spent blocked, from kymographs shown in **c–e** and Fig. 1c,l. Bars represent averaged means from three independent experiments. Error bars represent the s.e.m. **c–f**, Fields of view (left; scale bar: 2 μm) and kymographs (right; scale bars: 2 μm and 60 s) illustrating MT dynamics in indicated conditions. Orange arrowheads, blocked plus ends; white arrowheads, CEP104ΔTOG at seeds plus end; white arrows, CEP104-tracking minus ends. **g**, Parameters of MT plus-end dynamics, from kymographs shown in **f,h–l**, and Extended Data Fig. 2b. For dynamic state, bars represent the averaged means from three independent experiments. For depolymerization rate, bars represent the pooled data from three independent experiments: 2 nM CEP104, $n = 133$; 2 nM CEP104 with 15 nM CCDC66, $n = 23$; 10 nM CEP104 with 15 nM CCDC66, $n = 122$; 2 nM CEP104 with 10 nM CSPP1,

$n = 69$; 10 nM CEP104 with 10 nM CSPP1, $n = 99$; EB3 with 2 nM CEP104, $n = 1$; EB3 with 2 nM CEP104 and 15 nM CCDC66, $n = 63$; EB3 with 2 nM CEP104 and 10 nM CSPP1, $n = 116$. **h–n**, Fields of view (left; scale bar: 2 μm) and kymographs (right; scale bars: 2 μm and 60 s) illustrating MT dynamics in indicated conditions. Light-orange arrowheads, blocked seeds; dark-orange arrowhead, blocked lattice; blue arrowheads, slow plus-end depolymerization. **o**, Fields of view (left; scale bar: 2 μm) and kymographs (right; scale bars: 2 μm and 60 s) illustrating MT dynamics in indicated conditions. Orange arrowheads, blocked plus ends. **p**, Percentage of time MT plus ends spent blocked, from kymographs shown in **m–o** and Extended Data Fig. 2b. Bars represent the averaged means from three independent experiments. **q**, Scheme showing CEP104 domain functions and interactions with other CTM proteins. In **b,g,p**, error bars represent the s.e.m. ****$P < 0.0001$ (Kruskal–Wallis test followed by Dunn's post hoc test).

conditions, CEP104 and CCDC66 could also strongly inhibit depolymerization of dynamic MTs by either pausing the plus ends or reducing their shrinkage rate from $36.91 \pm 1.34$ μm min$^{-1}$ to $0.12 \pm 0.02$ μm min$^{-1}$ (Fig. 3g,h). Similar effects were observed when either 2 nM or 10 nM CEP104 was combined with 10 nM CSPP1; growth inhibition at the seed and at the dynamic plus ends and slow depolymerization were observed (Fig. 3g,i,j). The addition of EB3 had no strong impact on MT dynamics when combined with 2 nM CEP104 and 15 nM CCDC66 (Fig. 3f,g,k) but potentiated seed blocking when included with 2 nM CEP104 and 10 nM CSPP1, as most MTs were blocked at the seed in these conditions (Fig. 3g,i,l).

On the basis of our own and previously published coimmuno-precipitation experiments, the interactions of CCDC66 and CSPP1 with CEP104 depended on its jelly-roll (JR) domain (Extended Data Fig. 2c)[15,20]. The deletion of this domain abolished the ability of both proteins to recruit CEP104 to MTs (Fig. 3m,n and Extended Data Fig. 2d). CEP104ΔJR could still block seed outgrowth in the presence of EB3 (Fig. 3o,p), although a higher concentration of CEP104ΔJR compared to full-length CEP104 was needed to achieve significant plus-end blocking (50 nM versus 10 nM; compare Fig. 1d,l to Fig. 3o,p and Extended Data Fig. 2b), indicating that the JR domain might directly or indirectly contribute to MT blocking.

Our results demonstrate that CEP104 can stably and specifically associate with MT plus ends and inhibit both tubulin addition and removal from these ends in a TOG-domain-dependent manner. However, the affinity of CEP104 for MTs is rather low and its activity is strongly enhanced by association with its binding partners EB3, CCDC66 and CSPP1 that recruit it to MTs. Because EB3 binds to the outer MT surface[47] and the same might be true for CCDC66 as it can bundle MTs[23], whereas CSPP1 associates with the MT lumen[34], CEP104 might be positioned at protofilament ends to allow access to both MT surfaces (Fig. 3q).

## TOGARAM1-induced rescues depend on TOG3 and TOG4 domains

Next, we turned to TOGARAM1, a protein that contains two pairs of TOG domains (TOG1–TOG4) connected by a long linker (Fig. 4a). Single-molecule counting experiments showed that it is a monomer

(Fig. 4b), similar to other MT regulators with multiple TOG domains, XMAP215 (ref. 45) and CLASP2 (ref. 48). A previous study showed that isolated TOG2 and TOG4 but not TOG1 and TOG3 promoted tubulin polymerization in light-scattering assays[19]. To test how these domains contribute to the rescue activity of TOGARAM1, we took advantage of previously generated mutants where the tubulin-binding surfaces of the TOG domains were disrupted by substitutions within the intra-HEAT loop of HEAT repeat A[19] (Fig. 4a). Although there is no direct interaction between TOGARAM1 and EB3, we included EB3 in our in vitro assays as EB3 increases catastrophe frequency and, therefore, facilitates observing rescues (Fig. 1e)[42]. We found that simultaneous mutations impacting TOG3 and TOG4 (TOGARAM1 123'4') but not TOG1 and TOG2 (TOGARAM1 1'2'34) abolished the rescue activity, although both mutants could still bind to MTs at 10 nM concentration (Fig. 4a,c–e and Extended Data Fig. 3). This was different from previous observations in cells, where the double TOG3–TOG4 mutant (TOGARAM1 123'4') could not bind to MTs[19]. TOG3 and TOG4 showed redundancy within the full-length protein because single mutations impacting either of them did not abolish rescues (Fig. 4a,c,f,g and Extended Data Fig. 3). TOG1 and TOG2 could be deleted without impairing rescue activity (Fig. 4a,c,h and Extended Data Fig. 3) and the combination of these two domains (TOG1–TOG2) without the adjacent linker region did not bind to MTs even at 200 nM (Fig. 4a,i and Extended Data Fig. 3). TOG3–TOG4 did not visibly associate with MTs at 10 nM but, at 200 nM, it bound to MTs and induced rescues (Fig. 4a,c,j,k and Extended Data Fig. 3). Because the construct containing TOG3–TOG4 domains together with adjacent linker (L-TOG-34) was active in our assays already at 10 nM, the linker region of TOGARAM1 contributes to MT affinity in agreement with previous findings[19]. All tested constructs mildly reduced MT growth rate and TOG3–TOG4 constructs with or without the adjacent linker mildly suppressed catastrophes (Fig. 4c). Altogether, on its own, TOGARAM1 has relatively mild effects on MT dynamics, which rely on its two C-terminal TOG domains.

## ARMC9 enhances the effects of TOGARAM1 and CSPP1

We next set out to characterize the function of ARMC9. SEC revealed that ARMC9 forms multimers (Extended Data Fig. 1b) and single-molecule counting experiments showed that ARMC9 forms dimers dependent

**Fig. 4 | The rescue activity of TOGARAM1 depends on the TOG3 and TOG4 domains. a**, Schematic representation of different TOGARAM1 constructs and summary table highlighting their effects on MT dynamics. Vertical lines indicate point substitutions predicted to ablate tubulin-binding activity. **b**, Fields of view (top) and histogram plot (bottom) of fluorescence intensities of single GFP molecules, EB3 dimers and TOGARAM1 molecules immobilized in separate chambers of the same coverslip. Number of molecules analyzed: GFP, $n = 57{,}865$; EB3, $n = 73{,}074$; TOGARAM1, $n = 74{,}306$. Scale bar: 2 μm. **c**, Parameters of MT plus-end dynamics in the presence of 20 nM EB3 in combination with indicated concentrations of TOGARAM1 constructs (from kymographs shown in **d–h,k** and Fig. 1c,p). For growth rates, bars represent the pooled data from three independent experiments. Total number of growth events: EB3 alone,

$n = 938$; EB3 with TOGARAM1, $n = 861$; EB3 with TOMOGRAM1 1'2'34, $n = 350$; EB3 with TOMOGRAM1 123'4', $n = 337$; EB3 with TOMOGRAM1 123'4', $n = 560$; EB3 with TOMOGRAM1 1234', $n = 253$; EB3 with L-TOG-34, $n = 131$; EB3 with 200 nM TOG3–TOG4, $n = 150$. For transition frequencies, bars represent the averaged means from three independent experiments. Error bars represent the s.e.m., ****$P < 0.0001$ (Kruskal–Wallis test followed by Dunn's post hoc test). **d–k**, Fields of view (left; scale bar: 2 μm) and kymographs (right; scale bars: 2 μm and 60 s) illustrating MT dynamics from GMPCPP-stabilized seeds with 20 nM GFP–EB3 or mCherry–EB3 and indicated concentrations and colors of TOGARAM1 constructs. Assays were repeated three independent times. Blue arrowheads, rescues.

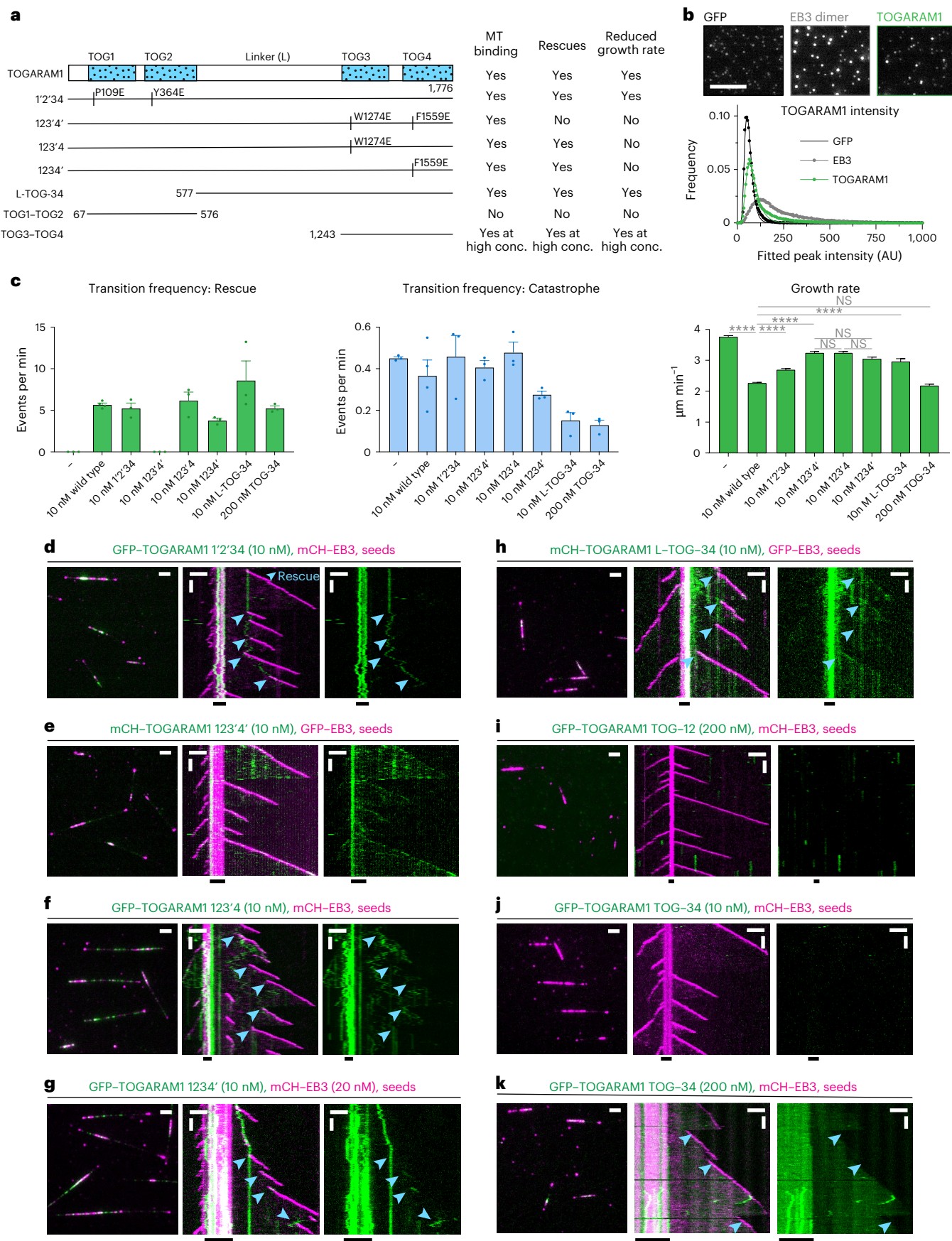

on its central H region (Extended Data Fig. 4a,b). ARMC9 does not bind to MTs on its own or in the presence of CEP104 and EB3 (Fig. 1g,h and Extended Data Fig. 2b). Consistently, no direct interaction has been described for ARMC9 and CEP104, even though they were previously shown to weakly coimmunoprecipitate with each other[9], an observation we could not confirm (Extended Data Fig. 4c). We confirmed by coimmunoprecipitation the interaction between ARMC9 and CCDC66; however, we did not see any recruitment of ARMC9 by CCDC66 to the MT lattice in reconstitution experiments (Extended Data Fig. 4c,d). We also confirmed by coimmunoprecipitation the interaction between ARMC9 and TOGARAM1 (Extended Data Fig. 4c) and refined the previous mapping of this interaction[9] by showing that it requires the ARM repeats of ARMC9 (but not its H domain responsible for dimerization) and the tubulin-binding surface of the TOG2 domain of TOGARAM1 (Fig. 4a and Extended Data Fig. 4a,e,f). We also tested the impact of different point substitutions identified in persons with Joubert syndrome on the interaction of ARMC9 and TOGARAM1 (refs. 9,10) and found that all tested substitutions in the TOG2 domain (all located in the first HEAT repeat) and one of the known substitutions in the ARM domain (G492R) perturb the binding (Extended Data Fig. 4a,g,h), supporting the functional importance of this interaction.

TOGARAM1 could recruit ARMC9 to MTs and the colocalization between the two proteins was particularly prominent in particles that could stationarily bind to or diffuse on MTs (Fig. 5a). As expected from the coimmunoprecipitation experiments, this colocalization was not perturbed by the 123'4' mutant of TOGARAM1 (Figs. 4a and 5b) but was abolished by the 1'2'34 mutant (Figs. 4a and 5c). The brightness of ARMC9 and TOGARAM1 in MT-bound particles increased over time, indicating binding of additional molecules (Fig. 5a,b). This binding was likely driven by ARMC9 as it was not observed for TOGARAM1 alone (Fig. 1p). The presence of ARMC9 had little effect on MT growth rates but did cause a roughly twofold increase in the rescue activity of TOGARAM1, which was still dependent on TOG3 and TOG4 (Fig. 5a,b,d). In contrast to ARMC9 and TOGARAM1, no colocalization between CCDC66 and TOGARAM1 was observed (Extended Data Fig. 4d).

We also observed coimmunoprecipitation of ARMC9 and CSPP1 (Extended Data Fig. 5a,b), confirming previous observations[9]. We mapped this interaction to the C-terminal part of CSPP1 and the linker regions surrounding the central H domain of ARMC9 (Extended Data Fig. 4a). CSPP1 could trigger accumulation of ARMC9 on MTs in the vicinity of the growing MT ends (Fig. 5e) and ARMC9 mildly increased the percentage of time CSPP1 induced MT pausing despite the average pause duration being shorter (Fig. 5f). The C-terminal portion of CSPP1 was required to recruit ARMC9 to MTs, confirming our coimmunoprecipitation experiments (Fig. 5g,h and Extended Data Fig. 5a). As previously reported, the shorter CSPP-S construct was less potent at pausing MTs, whereas the CSPP-MT-organizing region induced relatively few pauses but promoted rescue events (Extended Data Fig. 5c)[15,17,34] and these effects were not changed by the addition of ARMC9 (Fig. 5g,h).

We also tested whether CSPP1 and TOGARAM1 can bind to each other but detected no coimmunoprecipitation (Extended Data Fig. 5d) and CSPP1 displayed no colocalization with either TOGARAM1 or CCDC66 in the in vitro assays (Extended Data Fig. 5e). Altogether, our results indicate that CCDC66 does not colocalize with either TOGARAM1 or CSPP1 in reconstitution assays, whereas ARMC9 can bind to both proteins through two distinct domains and mildly enhance their effects on MT dynamics (Fig. 5i).

## Combination of CTM proteins drives slow processive MT growth

To complete the exploration of all pairwise combinations of CTM proteins, we examined the joint effects of CEP104 and TOGARAM1. The two proteins coprecipitated with each other in a manner dependent on the zinc finger (ZNF) domain of CEP104 and the linker region of TOGARAM1 (Figs. 3a and 4a and Extended Data Fig. 6a,b). When CEP104 and TOGARAM1 were combined on dynamic MTs in the presence of EB3, CEP104 was located not only at the plus ends but also at MT seeds, likely because of the binding of CEP104 to TOGARAM1, which is enriched at MT seeds (Figs. 1p and 6a and Extended Data Fig. 6c). Together, CEP104 and TOGARAM1 induced periods of pausing at the plus ends of both seeds and dynamic MTs (Fig. 6a–c and Extended Data Fig. 6c). However, these pauses were transient and followed by growth events (Fig. 6c–e), something we did not see with any other CEP104–ciliary tip protein combination. Furthermore, the two proteins together induced periods of very slow MT polymerization with the rates of $0.12 \pm 0.01$ µm min$^{-1}$, ~20 times slower than the $2.27 \pm 0.02$ µm min$^{-1}$ observed with TOGARAM1 alone (Fig. 6a,c,d). CEP104 and TOGARAM1 can, thus, interact through regions that do not engage their tubulin-binding domains and together impose slow MT polymerization (Fig. 6f).

Next, we combined all five CTM components in the presence of EB3. Strikingly, in these conditions, all MTs displayed slow and highly processive plus-end polymerization with occasional pausing, with an average rate of $0.19 \pm 0.04$ µm min$^{-1}$, while minus ends could still undergo phases of growth and shrinkage (Fig. 7a–d and Extended Data Fig. 6d,e). Unlike in the conditions where CEP104 blocked MT plus ends (Fig. 2c), FRAP of CEP104 on slowly growing MT ends showed partial, albeit very slow, recovery over the course of 10 min, indicating that some turnover of CEP104 is needed for tubulin addition (Extended Data Fig. 6f). To test which proteins are essential for slow growth, we repeated the assays while leaving out each of the CTM proteins. Omitting TOGARAM1 abolished slow growth completely as all MT plus ends were paused (Fig. 7a,d,e). This indicates that an important function of TOGARAM1 is to overcome MT growth inhibition imposed by two MT-pausing factors, CEP104 or CSPP1. Leaving out CEP104 had no major effect, except for a slight increase in pausing and very occasional periods of fast growth (Fig. 7a,c,d,f), whereas fast elongation became much more common in the absence of CSPP1 (Fig. 7a,c,d,g). Omitting

**Fig. 5 | ARMC9 colocalizes with TOGARAM1 and CSPP1 and enhances their effects on MT dynamics. a–c**, Fields of view (left; scale bar: 2 µm) and kymographs (right; scale bars: 2 µm and 60 s) illustrating MT dynamics from GMPCPP-stabilized seeds with 20 nM GFP–EB3 or mCherry–EB3 in indicated conditions. Assays were repeated independently three times. Blue arrowheads, ARMC9 and TOGARAM1 colocalization. **d**, Parameters of MT plus-end dynamics in the presence of 20 nM EB3 in combination with indicated concentrations of TOGARAM1 constructs and ARMC9 (from kymographs shown in **a,b** and Figs. 1c,p and 4e). For growth rates, bars represent the pooled data from three independent experiments. Total number of growth events: EB3 alone, n = 938; EB3 with TOGARAM1, n = 861; EB3 with TOGARAM1 and ARMC9, n = 342; EB3 with TOGARAM1 123'4', n = 337; EB3 with TOGARAM1 123'4' and ARMC9, n = 325. For transition frequencies, bars represent averaged means from three independent experiments. Error bars represent the s.e.m. ****P < 0.0001 (Kruskal–Wallis test followed by Dunn's post hoc test). **e**, Fields of view (left; scale bar: 2 µm) and kymographs (right; scale bars: 2 µm and 60 s) illustrating MT dynamics from GMPCPP-stabilized seeds with 20 nM GFP–EB3 and CTM proteins indicated. Yellow arrowheads, colocalization. **f**, Parameters of MT plus-end dynamics with 20 nM EB3 in combination with indicated concentrations of ciliary top module proteins (from kymographs shown in **e** and Fig. 1c,n). For growth rate and pause duration, bars represent the pooled data from three independent experiments. Total number of growth events, pauses: EB3 alone, n = 938, 0; EB3 with CSPP1, n = 1,715, 273; EB3 with CSPP1 and ARMC9, n = 627, 315. For transition frequencies and dynamic state, bars represent the averaged means from three independent experiments. Error bars represent the s.e.m. ****P < 0.0001 (Kruskal–Wallis test followed by Dunn's post hoc test). **g,h**, Fields of view (left; scale bar: 2 µm) and kymographs (right; scale bars: 2 µm and 60 s) illustrating MT dynamics from GMPCPP-stabilized seeds with 20 nM GFP–EB3 in indicated conditions. Assays were repeated independently three times. Yellow arrowheads, colocalization. **i**, Schematic model showing TOGARAM1, ARMC9 and CSPP1 interaction domains.

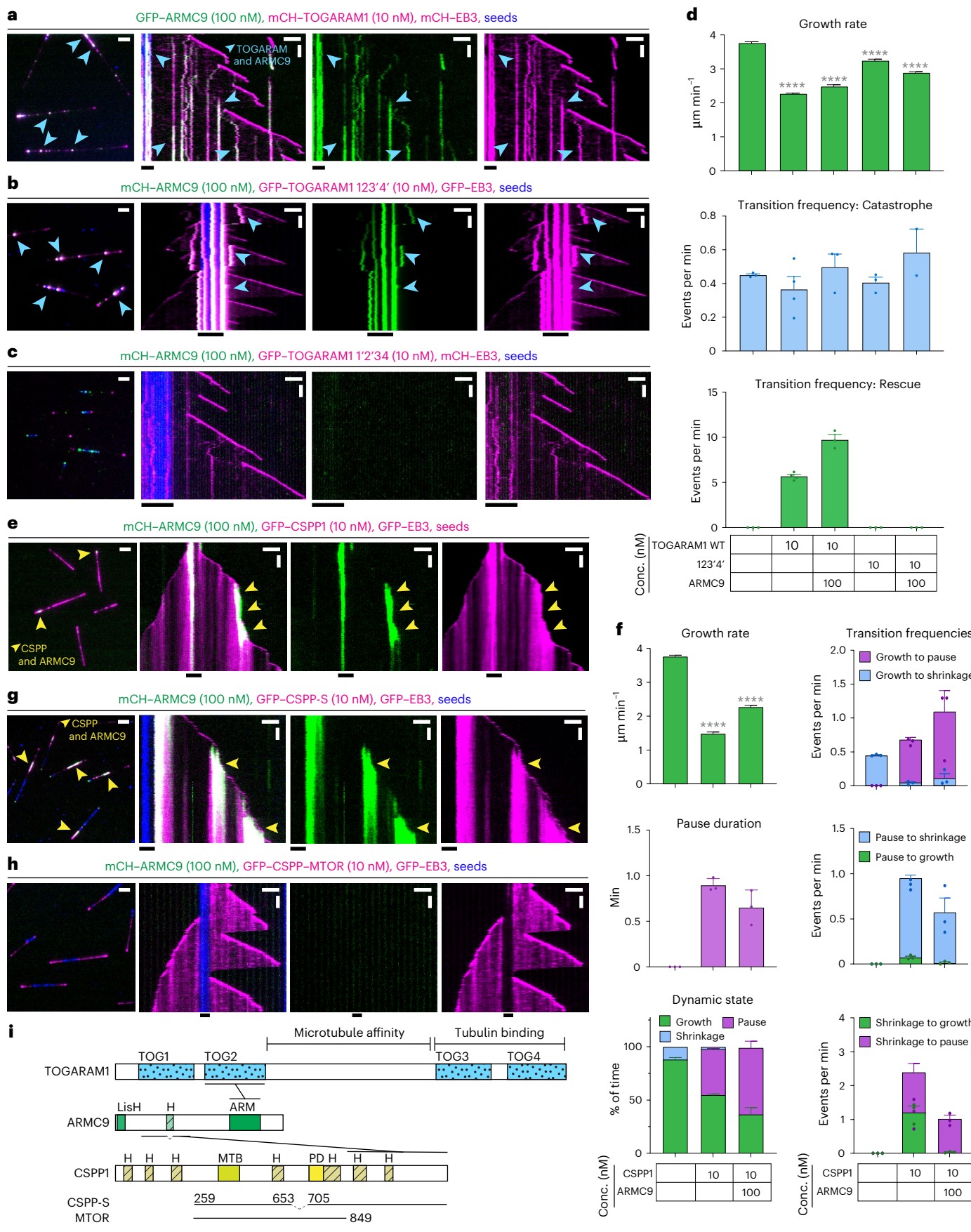

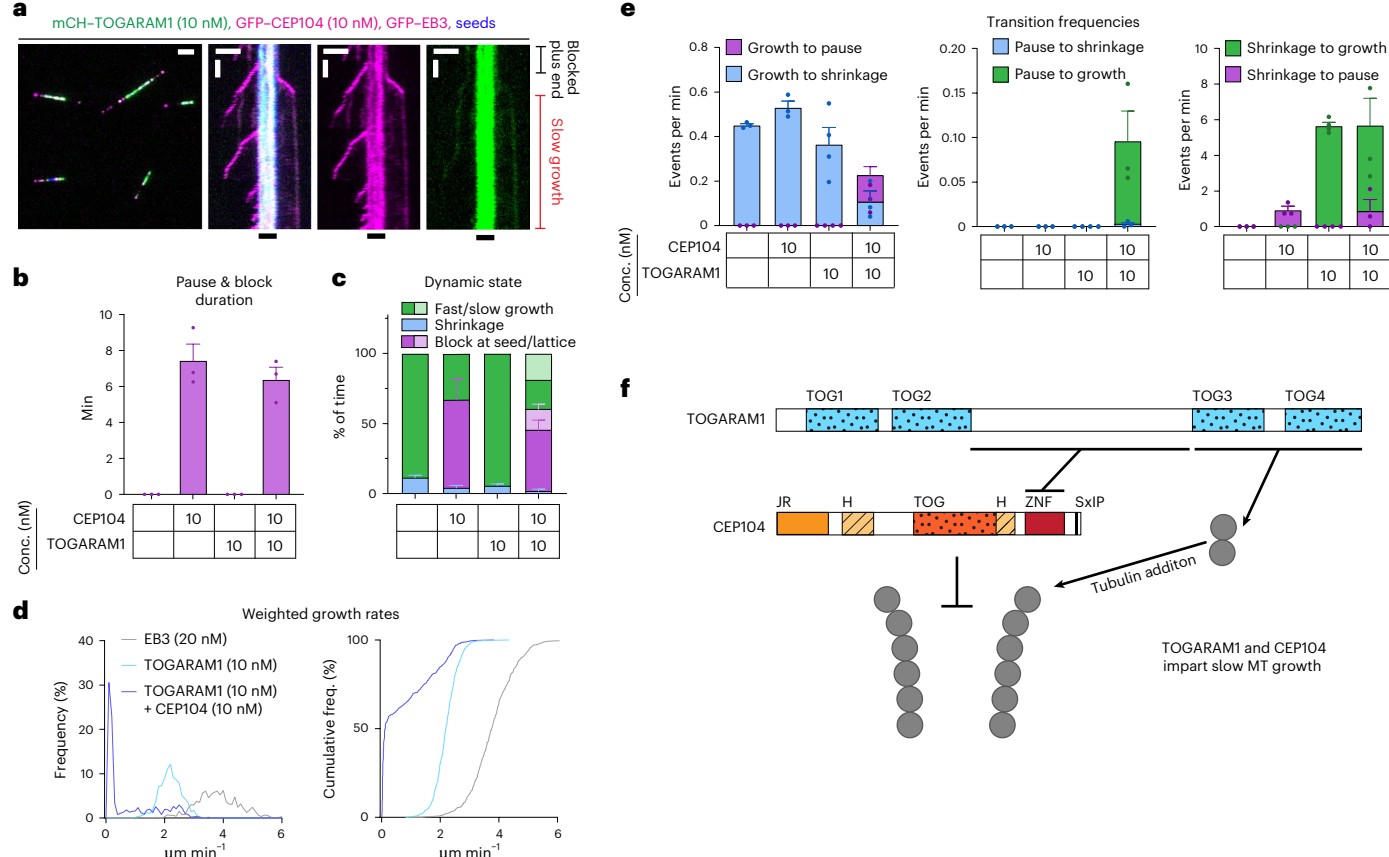

**Fig. 6 | CEP104 and TOGARAM1 are sufficient to impart slow MT growth. a**, Field of view (left; scale bar: 2 µm) and kymograph (right; scale bars: 2 µm and 60 s) illustrating MT dynamics with GFP–EB3, mCherry–TOGARAM1 and GFP–CEP104. **b**–**e**, Parameters of MT plus-end dynamics in the presence of EB3 in combination with indicated concentrations of CTM proteins (from kymographs shown in **a** and Fig. 1c,l,p). For pause and block duration (**b**) and growth rates (**d**), data were pooled from three independent experiments. Total number of

growth events, pauses: EB3 alone, *n* = 938, 0; EB3 with CEP104, *n* = 213, 101; EB3 with TOGARAM1, *n* = 861, 0; EB3 with CEP104 and TOGARAM1, *n* = 373, 140. Bars represent the mean ± s.e.m. For dynamic state (**c**) and transition frequencies (**e**), bars represent the averaged means from three independent experiments. Error bars represent the s.e.m. **f**, Scheme showing interactions between CEP104, TOGARAM1 and tubulin.

either ARMC9 or CCDC66 had no major effect except that no pausing was observed and occasional episodes of fast growth were detected (Fig. 7a,c,d,h,i). No MT catastrophes or shrinkage were observed in any of these conditions (Fig. 7d).

Next, we removed ARMC9 and CCDC66 simultaneously and were surprised to see that, in these conditions, with TOGARAM1, CEP104 and CSPP1 at 10 nM each, almost all MT plus ends were paused (Fig. 7d and Extended Data Fig. 6g). However, when we reduced the concentration of CEP104 and CSPP1 to 2 nM in the presence of 10 nM TOGARAM1, episodes of slow growth with an average rate of 0.022 ± 0.01 µm min⁻¹ were again observed (Fig. 7d,j and Extended Data Fig. 6h). While raising TOGARAM1 concentration to 20 nM further increased the percentage of the time MTs spent growing slowly, the rate of these slow growth episodes remained essentially the same (Fig. 7d,j and Extended Data Fig. 6i). The slow growth rate also remained relatively constant when the concentration of either CEP104 or CSPP1 was decreased to 0.5 nM but periods of fast growth were again observed in these conditions (Fig. 7d,j and Extended Data Fig. 6j,k). Together, these results demonstrate that slow growth is a relatively robust state triggered at a certain ratio of TOGARAM1 and one or two pausing factors. When the latter are present in excess, the scaffold proteins ARMC9 and CCDC66 can stimulate TOGARAM1-dependent slow growth. Altogether, CTM components stabilize MTs by preventing their depolymerization and promoting robust but very slow tubulin addition at a rather constant rate.

## CTM proteins form cork-like densities at MT plus ends

To better understand CTM localization on MT ends, we turned to cryo-ET. We reconstructed three-dimensional (3D) volumes containing MTs grown with either EB3 alone or in the presence of the entire CTM. On the basis of our analysis of MT dynamics, we presumed that MTs were mostly fast growing for EB3 controls and elongating slowly with a few paused ends for CTM samples (Fig. 7d). MTs grown in the presence of the CTM showed densities at their ends reminiscent of champagne corks, whereas MTs grown in the presence of only EB3 had no clear density at their ends (Fig. 8a,b,d,e and Extended Data Fig. 7a,b). Despite the density for the CTM appearing quite varied in structure, it was typically situated both in the lumen and on the tips of the MT protofilaments (Fig. 8b,e and Extended Data Fig. 7b). We used two-dimensional (2D) rotational averaging to resolve MT polarity on the basis of the chirality of the cross-section[49], through which we determined that cork-like electron densities were specific for MT plus ends (Fig. 8a–c and Extended Data Fig. 7b). Additionally, 85% of plus ends found in the CTM dataset, where the chirality of the cross-section could be resolved, were corked. Furthermore, all minus ends were free (Fig. 8c). At corked MT tips, extensive contacts on the luminal surface of tubulin and the exposed protofilament ends were observed. Luminal densities were also present at regions located away from the cork; however, we could not detect any visible electron densities at the outer MT surface (Extended Data Fig. 7c,d).

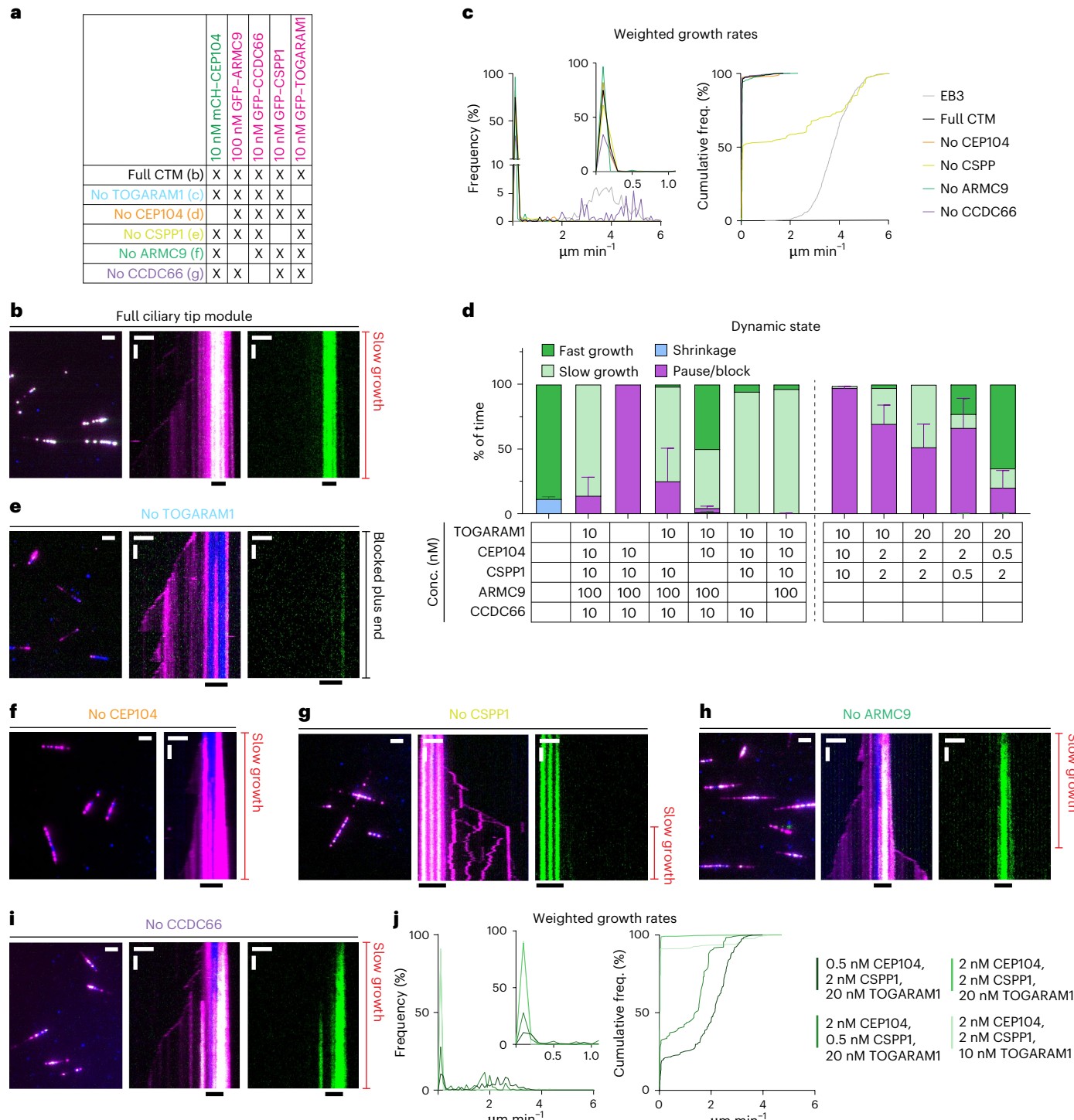

**Fig. 7 | Combined action of CTM proteins drives slow processive MT growth.**
**a,b**, Reconstitutions of indicated concentrations and colors of CTM proteins (**a**). Fields of view (left; scale bar: 2 μm) and kymograph (middle; combined colors; right, CEP104 only; scale bars: 2 μm and 60 s) (**b**) illustrating MT dynamics from GMPCPP-stabilized seeds with 20 nM GFP–EB3 (in magenta) and CTM components indicated in **a**. **c,d**, Weighted growth rates (**c**) and dynamic state (**d**) for combinations of CTM proteins indicated (from kymographs in **b,e**–**i**, Fig. 1c and Extended Data Fig. 6e–k). Growth rates in **c** are represented as histograms (left; with the inset showing enlargement for the rate values under 1 μm min⁻¹) and cumulative frequency diagrams (right). Data were pooled from three independent experiments. Total number of growth events: EB3 alone, *n* = 938; entire CTM, n = 70; no ARMC9, *n* = 81; no CCDC66, *n* = 83; no CEP104, *n* = 41; no

CSPP1, *n* = 141. For dynamic state (**d**), bars represent the averaged means from two independent experiments and error bars represent the s.e.m. **e**–**i**, Fields of view (left; scale bar: 2 μm) and kymographs (middle, combined colors; right, CEP104 only; scale bars: 2 μm and 60 s) illustrating MT dynamics from GMPCPP-stabilized seeds with 20 nM GFP–EB3 (magenta) and CTM components indicated in **a**. **j**, Weighted growth rates (plotted as in **c**) for combinations of indicated CTM proteins (from kymographs in Extended Data Fig. 6h–k). Data were pooled from three independent experiments. Total number of growth events: 2 nM CEP104 with 2 nM CSPP1 and 10 nM TOGARAM1, *n* = 58; 2 nM CEP104 with 2 nM CSPP1 and 20 nM TOGARAM1, *n* = 88; 2 nM CEP104 with 0.5 nM CSPP1 and 20 nM TOGARAM1, *n* = 77; 0.5 nM CEP104 with 2 nM CSPP1 and 20 nM TOGARAM1, *n* = 228.

To visualize MT protofilaments and CTM, we initially performed semiautomated segmentation of the tomographic volumes (Fig. 8d,e). However, using this approach, it was difficult to distinguish individual protofilaments for further analysis. We, therefore, turned to manual protofilament tracing (see Methods) and were able to trace the majority (~13) of protofilaments of each MT end (Fig. 8f,g and Extended Data Fig. 7a,b). We measured both protofilament length (length of curved segment) and the s.d. of the distance between the first point that deviated from being completely straight to the end of the protofilament (Fig. 8h–j). MT plus ends grown with only EB3 showed primarily flared protofilaments[50,51] (Fig. 8a,f,h,i and Extended Data Fig. 7a), whereas plus ends 'corked' with CTM proteins displayed a dramatic decrease in both protofilament length and s.d. of protofilament length per MT (Fig. 8b,g,h,i and Extended Data Fig. 7b), as well as a mild decrease in average local curvature (Fig. 8k). Lastly, MT raggedness (the s.d. of the axial distance along the MT of the first point for each protofilament that deviated from being completely straight; Fig. 8j) was modestly but significantly decreased in corked MT plus ends (Fig. 8l). Our results indicate that the CTM acts from both the lumen and the tips of MTs to stabilize slow and processive elongation (Fig. 8m).

## Discussion

Axonemal MTs are stable and elongate very slowly without undergoing long depolymerization events. Here, we reconstituted these behaviors in vitro with five CTM components. We found that, collectively, these proteins impose very slow and processive MT growth. The slow growth state of MT plus ends was robust as, interestingly, we could not modulate its rate by altering protein concentrations, suggesting that it is not overly sensitive to protein stoichiometry. Furthermore, elongation rates measured in our assays, $0.19 \pm 0.04$ µm min$^{-1}$, fall within the range of those measured for the initial elongation of regenerating flagella in single-celled organisms such as *Chlamydomonas* (0.08–0.40 µm min$^{-1}$)[35–37,52], although these rates are reduced when flagella reach their normal length. It remains to be determined whether our measured growth rates are within the range of elongating primary cilia.

Slow growth is an unusual state for dynamic MTs because it is incompatible with the formation of a long stabilizing GTP cap needed to avoid MT disassembly[7] and promote growth[8]. Factors controlling slow MT polymerization must, thus, inhibit tubulin detachment and promote tubulin addition at MT plus ends that have only a very short or no GTP cap. We found that these functions require a combination of CTM proteins. Two of them, TOGARAM1 and CEP104, were predicted to be MT polymerases because their individual TOG domains can bind tubulin and enhance tubulin polymerization in vitro[19,30–32]. However, full-length TOGARAM1 and CEP104 do not accelerate tubulin polymerization at freely growing MT plus ends. On the contrary, TOGARAM1 somewhat slows down MT growth rate and CEP104 potently blocks MT elongation. Nevertheless, these proteins are MT stabilizers rather than depolymerases as both can inhibit tubulin detachment from GDP-bound MT ends; TOGARAM1 induces rescues while CEP104 dramatically slows down disassembly of MT plus ends. Moreover, TOGARAM1 can promote tubulin addition to paused MT ends and, together, CEP104 and TOGARAM1 can induce phases of very slow polymerization. TOGARAM1 can, thus, be regarded as a slow MT polymerase, which shows some functional similarities to CLASPs, TOG-domain proteins that mildly inhibit growth rate but potently promote regrowth of MT ends that start losing their GTP cap[48].

Slow processive MT growth in the presence of either CEP104 and TOGARAM1 combination or the whole CTM is plus end specific. This is likely driven by the specific orientation of the TOG domains, which can position tubulin dimers favorably for binding to protofilament tips at the plus end[53]. Recruitment of cytoplasmic TOG-domain proteins to MTs is known to require positively charged intrinsically disordered regions[48,54,55] and additional adaptors, such as EB1 or SLAIN2 (refs. 56,57). This is also true for the ciliary TOG-domain regulators. TOGARAM1, similar to XMAP215 (ref. 54), contains a linker region that promotes binding to MTs in vitro and is further recruited to MTs by ARMC9. MT binding by CEP104 is promoted by its partners, EB3, CSPP1 and CCDC66. EB3 is known to bind to the outer MT surface[47], whereas CSPP1 is a luminal protein[34]. CCDC66 is likely to be at least partly exposed on MT surface as it can bundle MTs in vitro[23]. With binding partners on both sides of the MT wall, CEP104 is in a good position to span the MT tip. TOGARAM1 is also likely positioned very close to protofilament ends because it can catalyze tubulin addition. ARMC9 promotes both TOGARAM1 activity and the pausing behavior of the luminal protein CSPP1 and could be located close to the inner surface of the MT plus end. Collectively, this fits with the cryo-ET data, which show that the complex of CTM proteins localizes at the luminal side (likely CSPP1) and the tip (likely CEP104 and TOGARAM1) of MT plus ends, resembling a cork on a champagne bottle.

The interactions between CTM proteins depend on nonoverlapping domains. For example, CEP104 binds to tubulin through its TOG domain, to EB3 through the SxIP motif within its flexible C-terminal region, to TOGARAM1 through the ZNF domain and to CCDC66 and CSPP1 through the JR domain (Figs. 3q and 6f). Similarly, TOGARAM1 interacts with tubulin through TOG3 and TOG4, with ARMC9 through TOG2 and with MTs and CEP104 through its linker region (Fig. 5i). For several reasons, we do not think that the five CTM proteins form a stoichiometric complex but rather a flexible interaction network, the function of which is to enhance the accumulation of these proteins at the distal ends of cilia. First, photobleaching experiments revealed that CEP104 exchanges, albeit slowly, in the context of the CTM. Second, the slow growth state can be achieved at different stoichiometries between the module components. Third, a flexible interaction network explains why we saw variation in size of corks in our cryo-ET data. Additionally, in vitro, CTM members act in a partially redundant fashion and it is possible that not all CTM proteins are present in every cork. Leaving individual CTM proteins out of reconstitutions while making

---

**Fig. 8 | CTM proteins form cork-like densities at MT plus ends and reduce protofilament flaring. a,b**, Left, slices (4.3 nm thick) of denoised tomograms showing MT plus ends grown in presence of EB3 alone (**a**) or CTM and EB3 (**b**). Right, an 8-nm-thick transverse cross-section of the corresponding MTs. The transverse cross-section (top right) is accompanied by its rotational average (bottom right) to indicate MT polarity. Scale bars: 25 nm. Independent experiments were run twice for controls and three time for CTM samples. **c**, Cork distribution in the CTM samples based on visual inspection for the presence of cork densities and based on chirality of the rotationally averaged cross-section for MT polarity. Total number of MT ends analyzed: plus ends, $n = 60$; minus ends, $n = 38$. **d,e**, The 3D rendered segmentation volumes of a free or dynamic MT plus end (**d**) and CTM-corked plus end (**e**). MT, gray; CTM density, green. Scale bars: 25 nm. **f,g**, A 3D model of manually traced protofilament shapes, accompanied by their transverse cross-sections. Scale bars: 25 nm. **h**, Distribution of protofilament length measured from the last segment within the MT wall until the tip of the protofilament. Total number of protofilaments analyzed: EB3, $n = 329$; CTM, $n = 441$. Lines represent the median (solid) and quartiles (dotted). ****$P < 0.0001$ (two-sided Mann–Whitney test). **i**, Protofilament length s.d. per MT. Total number of MTs analyzed: EB3, $n = 26$; CTM, $n = 33$. Bars represent the mean and error bars represent the s.e.m. ****$P < 0.0001$ (two-sided Mann–Whitney test). **j**, Schematic representation of parameters that were obtained from manual tracing of MT plus-end protofilaments. **k**, Average local protofilament curvature per MT. Total number of MTs analyzed: EB3, $n = 25$; CTM, $n = 32$. Bars represent the mean and error bars represent the s.e.m. **$P = 0.0061$ (two-sided Mann–Whitney test). **l**, MT raggedness plotted as s.d. of protofilament axial distance of last segment within the wall per MT. Total number of MTs analyzed: EB3, $n = 26$; CTM, $n = 33$. Bars represent the mean and error bars represent the s.e.m. **$P = 0.002$ (two-sided Mann–Whitney test). **m**, Schematic model of the CTM at an MT plus end.

slow growth less robust had no major effect on MT dynamics. The one exception was TOGARAM1, which was essential for MT elongation; this aligned well with the observation that overexpression of TOGARAM1 induces longer cilia[9]. Furthermore, in vertebrate cells, the loss of any

of the CTM proteins results in the generally mild phenotype of shorter cilia, whereas simultaneous loss of several module components has more severe consequences[9,15,19,20,32]. Lastly, a loose interaction mode is also in line with the fact that in other species, such as ciliates, these

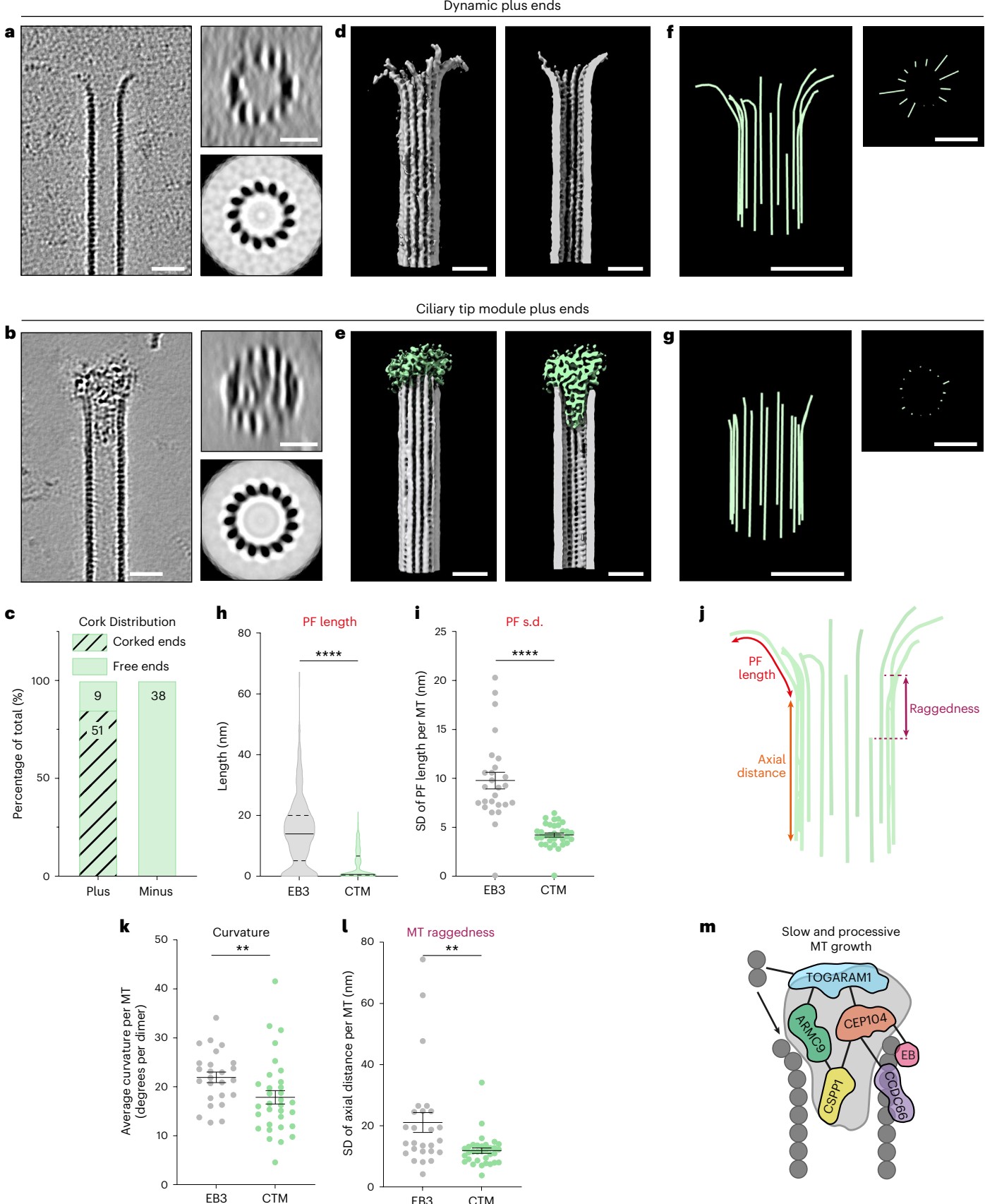

proteins bind to different sites (for example, A-tubules and B-tubules) and can have opposing functions[27].

It appears counterintuitive that deletions or mutations impacting all vertebrate CTM proteins cause shorter cilia even though CEP104 and CSPP1 inhibit MT elongation[9,15,19,20,32]. This could be explained by the interplay with additional factors not included in our assays. For example, CEP104 and CSPP1 might counteract MT-depolymerizing factors, such as kinesin 13, kinesin 18 KIF19A or kinesin 4 KIF7, which are known to act at ciliary tips[58–60]. Alternatively, CEP104 and CSPP1 might also cooperate with MT polymerization-promoting factors such as XMAP215 or CLASP. This idea is supported by CEP104 promoting centriole elongation in flies[24], where it is established that Orbit/CLASP and kinesin 13 KLP10A act as positive and negative regulators of fly centriole length, respectively[61]. Furthermore, our observation that both ARMC9 and CCDC66 can potentiate slow MT growth when the pausing factors CEP104 and CSPP1 predominate over TOGARAM1 helps to explain how scaffolding proteins can promote cilia elongation.

An important question is the nature of structural changes at MT plus ends associated with slow polymerization. Cryo-ET analysis showed that rapidly growing MTs terminate with strongly flared protofilaments[50,51], whereas cork-like structures formed by CTM proteins reduce protofilament flaring and at least partially cap the plus ends. Nevertheless, in these conditions, MT tips retain conformational variability, with some protofilaments more flared and some more blunt. It is possible that, at each given time, the protofilaments that are blunt are occluded by CTM proteins and, therefore, paused. Despite lacking a GTP cap, these protofilaments do not depolymerize because ciliary tip regulators keep them together and stabilize them from the luminal side. In this model, the more curved protofilaments are those undergoing tubulin addition, consistent with the idea that TOG domains increase local tubulin concentration and deliver 'curved' tubulins to protofilament tips[62]. Alternatively, CTM proteins could be responsible for keeping tubulin dimers within flared protofilaments in a bent state, while new dimers are added onto the straighter protofilaments, which slowly force the corks out of the plus ends as they elongate.

Formation of MT plus-end-specific corks that impose blunt protofilament conformation is strikingly similar to the 'plugs' observed in reconstitution experiments with the centriolar cap protein CP110 (ref. 63). MT stabilization from the luminal side combined with the reduction of protofilament curvature and occlusion of the longitudinal interface of β-tubulin might, thus, be a common mechanism for generation of very stable and slowly growing MTs, such as those of cilia and centrioles. Interestingly, although growth of centrioles and cilia is mostly controlled by distinct sets of proteins, CEP104 can bind to CP110 at the distal tip of the centriole[16,64,65] and has a role in centriole elongation in flies[24]. The relevance of luminal proteins in controlling the growth of cilia is supported by the observations of plugs at the tips of ciliary MTs in cells[39,66]. A mechanism involving the luminal MT surface and occlusion of the longitudinal interfaces of β-tubulin is different from the regulation of much more dynamic cytoplasmic MT ends, which involves factors located on the outer MT surface[67]. Strikingly, the fundamental principles are the same; in both cases, plus-end regulation depends on the balance of growth-promoting and growth-inhibiting activities that are brought together by a multivalent network of interacting proteins.

## Online content

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

## Methods

### DNA constructs, cell lines and cell culture

All proteins except CEP104 were cloned into modified pEGFP-C1 or pmCherry-C1 vectors with a StrepII tag and expressed in HEK293T cells (American Type Culture Collection, CRL-3216). CEP104 was cloned into pTT5-C1-based expression vectors (Addgene, 52355), which also had a StrepII tag and fluorescent tags (GFP or Cherry). For all module members except TOGARAM1, the human sequence was used; because of technical difficulties, the mouse sequence of TOGARAM1 was used, which is 84% identical to human TOGARAM1. Mouse TOGARAM1 constructs were a gift from K. Slep (The University of North Carolina, Chapel Hill). All CSPP1 constructs used were previously published[34]. HEK293T cells (American Type Culture Collection, CRL-3216) were cultured in DMEM:F10 (1:1) for coimmunoprecipitation assays or DMEM (Lonza) for protein purification; both were supplemented with 10% fetal calf serum (GE Healthcare Life Sciences) and 1% (v/v) penicillin–streptomycin. All cells were routinely checked for *Mycoplasma* contamination using the MycoAlertTM *Mycoplasma* detection kit (Lonza).

### Coimmunoprecipitation assays

HEK293T cells were transiently transfected with a mix consisting of polyethyleneimine (Polysciences) and constructs for the CTM proteins. Coimmunoprecipitation was performed with ChromoTek GFP-Trap magnetic particles M-270 (Proteintech) following the manufacturer's instructions with company advised buffers. Then, 24 h after transfection, cells were harvested in 1× PBS and lysed for 30 min on ice in 200 µl of lysis buffer (10 mM Tris-HCl pH 7.5, 150 mM NaCl, 0.5 mM EDTA and 0.5% IGEPAL CA-630) supplemented with EDTA-free protease inhibitor cocktail (Roche) and PhosSTOP phosphatase inhibitor cocktail (Roche). Lysate was cleared by 20-min centrifugation at 4 °C at 17,000$g$ and diluted with 300 µl of dilution buffer (10 mM Tris-HCl pH 7.5, 150 mM NaCl and 0.5 mM EDTA) supplemented with EDTA-free protease inhibitor cocktail (Roche) and PhosSTOP phosphatase inhibitor cocktail (Roche) per sample. Diluted lysate was incubated rotating end-over-end for 45 min with wash buffer equilibrated beads. Beads were washed four times with 500 µl of wash buffer (10 mM Tris-HCl pH 7.5, 150 mM NaCl, 0.5 mM EDTA and 0.05% IGEPAL CA-630) and protein was eluted in 80 µl of 2× Laemmli sample buffer (0.125 M Tris-HCl, 20% glycerol, 10% 2-mercaptoethanol, 0.02% bromophenol blue and 0.2 M DTT) and heated at 95 °C for 5 min. All steps before elution were performed at 4 °C and all reagents and samples were kept on ice.

Prepared samples were run on 8% SDS–PAGE gel and blotted onto Amersham Protran Premium 0.45-µm NC nitrocellulose membrane (Cytiva) by wet transfer at 37 V constant overnight at 4 °C in transfer buffer (0.2 M Tris-HCl, 2 M glycine, 10% methanol and 0.01% SDS) or by semidry transfer for 1 h at 0.15 A constant in transfer buffer (0.2 M Tris-HCl, 2 M glycine and 10% methanol). Membranes were blocked in blocking buffer (2% BSA diluted in 1× PBS and 0.05% Tween-20) for 1 h and sequentially incubated with primary antibodies and secondary antibodies diluted in blocking buffer for 1 h rotating at room temperature (primary and secondary antibodies) or overnight at 4 °C (primary antibodies only). Blots were thoroughly washed in between and after incubation with antibodies with wash buffer (1× PBS and 0.05% Tween-20) and imaged with an Odyssey CLx Infrared Imaging System (LI-COR biosciences) at various exposure times. The following antibodies were used for western blotting: rabbit anti-RFP (1:2,000; Rockland immunochemicals), mouse anti-GFP (1:2,000; Sigma-Aldrich), donkey anti-mouse IgG (H + L) IRDye® 800CW (1:10,000; LI-COR bioscience) and donkey anti-rabbit IgG (H + L) IRDye 680CW (1:10,000; LI-COR bioscience).

### Protein purification from HEK293T cells for in vitro reconstitution assays

HEK293T cells were transiently transfected with a mix consisting of polyethyleneimine (Polysciences) and constructs for one of the CTM proteins. The cells were harvested 28 h after transfection in lysis buffer

(50 mM HEPES, 300 mM NaCl, 1 mM MgCl₂, 1 mM DTT and 0.5% Triton X-100, pH 7.4) supplemented with protease inhibitors (Roche) and kept on ice for 15 min. After clearance of the debris by centrifugation, the supernatant was incubated with 20 µl of StrepTactin beads (GE Healthcare) for 45 min. After several washing steps, five times with a 300 mM salt wash buffer (50 mM HEPES, 300 mM NaCl, 1 mM MgCl₂, 1 mM EGTA, 1 mM DTT and 0.05% Triton X-100, pH 7.4) and three times with a 150 mM salt wash buffer (50 mM HEPES, 150 mM NaCl, 1 mM MgCl₂, 1 mM EGTA, 1 mM DTT and 0.05% Triton X-100, pH 7.4), the protein was eluted in elution buffer (50 mM HEPES, 150 mM NaCl, 1 mM MgCl₂, 1 mM EGTA, 1 mM DTT, 0.05% Triton X-100 and 2.5 mM *d*-desthiobiotin (Sigma-Aldrich), pH 7.4). Purified proteins were snap-frozen and stored at −80 °C.

### SEC

To ensure that purified proteins were not aggregating, SEC was performed at 4 °C using a Superose 6 increase 10/300 GL column (Cytiva). The ÄKTAmicro system was equilibrated with 50 mM HEPES, 150 mM NaCl, 1 mM MgCl₂, 1 mM EGTA and 0.05% Triton X-100 with a flow rate of 0.5 ml min⁻¹. For each experiment, 80–100 µl of protein sample containing 2–10 µg of protein was loaded into the column. As the priority with the SEC was to detect protein aggregation, the resin chosen was unable to reveal information about dimeric protein states.

### MS

To confirm each protein was purified and the eluted protein did not contain any contaminants or known interactors that could affect MT dynamics, we performed MS analysis. MS measurements were performed as described previously[34]. The purified protein sample was digested using S-TRAP microfilters (ProtiFi) according to the manufacturer's protocol. Digested peptides were eluted and dried in a vacuum centrifuge before liquid chromatography–MS (LC–MS) analysis. The samples were analyzed by reversed-phase nanoLC–MS/MS using an Ultimate 3000 ultrahigh-performance LC coupled to an Orbitrap Exploris 480 MS instrument (Thermo Fisher Scientific). Digested peptides were separated using a 50-cm reversed-phase column packed in-house (Agilent Poroshell EC-C18, 2.7 µm, 50 cm × 75 µm) and eluted from the column at a flow rate of 300 nl min⁻¹. MS data were acquired using a data-dependent acquisition method with an MS1 resolution of 60,000 and mass range of 375–1,600 $m/z$. Fragmentation spectra were collected at a resolution of 15,000 using a higher-energy collisional dissociation of 28, isolation window of 1.4 $m/z$ and fixed first mass of 120 $m/z$. MS/MS fragment spectra were searched using Sequest HT in Proteome Discoverer (Thermo Fisher Scientific) against a human database (UniProt, 2020) that was modified to contain the tagged protein sequence from each CTM protein and a common contaminant database. Tryptic miss cleavage tolerance was set to 2, fixed modifications were set to cysteine carbamidomethylation and variable modifications included oxidized methionine and protein N-terminal acetylation. Peptides that matched the common contaminate database were filtered out from the results table.

### In vitro reconstitution assays

**MT seed preparation.** Double-cycled GMPCPP-stabilized MT seeds were used as templates for MT nucleation in vitro assays. GMPCPP-stabilized MT seeds were prepared as described before[34]. A tubulin mix consisting of 70% unlabeled porcine brain tubulin, 18% biotin-labeled porcine tubulin and 12% HiLyte-488-labeled, rhodamine-labeled or HiLyte647-labeled porcine tubulin (all from Cytoskeleton) was incubated with 1 mM GMPCPP (Jena Biosciences) at 37 °C for 30 min. Polymerized MTs were then pelleted using an Airfuge for 5 min at 199,000$g$ and subsequently depolymerized on ice for 20 min. Next, 1 mM GMPCPP was added and MTs were let to polymerize again at 37 °C for 30 min. Polymerized MTs were again pelleted (as above) and diluted tenfold in MRB80 buffer containing 10% glycerol before snap-freezing to store them at −80 °C.

**In vitro reconstitution assays.** In vitro assays with dynamic MTs were performed as described before[34]. Microscopic slides were prepared by adding two strips of double-sided tape to mount plasma-cleaned glass coverslips. The coverslips were functionalized by sequential incubation with 0.2 mg ml⁻¹ PLL-PEG-biotin (Susos) and 1 mg ml⁻¹ neutravidin (Invitrogen) in MRB80 buffer (80 mM piperazine-N and N[prime]-bis(2-ethane sulfonic acid) pH 6.8, supplemented with 4 mM MgCl₂ and 1 mM EGTA). Then, GMPCPP-stabilized MT seeds were attached to the coverslips through biotin–neutravidin interactions. The coverslip was blocked with 1 mg ml⁻¹ κ-casein before the reaction mix was flushed in. The reaction mix consisted of different concentrations and combinations of fluorescently labeled purified proteins in MRB80 buffer supplemented with 15 µM porcine brain tubulin (100% dark porcine brain tubulin when 20 nM GFP–EB3 or mCherry–EB3 was added or 97% dark porcine brain tubulin when 3% rhodamine-labeled or HiLyte-488-labeled porcine tubulin was added), 0.1% methylcellulose, 1 mM GTP, 50 mM KCl, 0.2 mg ml⁻¹ κ-casein and oxygen scavenger mix (50 mM glucose, 400 µg ml⁻¹ glucose oxidase, 200 µg ml⁻¹ catalase and 4 mM DTT). This mix was spun down in an Airfuge for 5 min at 119,000g before addition to the flow chamber and the flow chamber was sealed with vacuum grease. MTs were imaged immediately at 30 °C using a TIRF microscope.

**In vitro assays for cryo-ET**
Samples were prepared as above for reconstitution assays with slight modifications. Instead of flow chambers, all steps occurred in a tube. Reaction mixes consisted of either just 20 nM EB3 or all CTM components (20 nM TOGARAM1, 20 nM CEP104, 20 nM CSPP1, 20 nM CCDC66 and 200 nM ARMC9) with 20 nM EB3 in MRB80 buffer supplemented with 15 µM porcine brain tubulin, 1 mM GTP, 0.2 mg ml⁻¹ κ-casein and 15 µM DTT. After centrifugation of the reaction mix for dynamic MTs, GMPCPP-stabilized seeds and ProtA-Au⁵ fiducials (1:20) were added and MTs were left to polymerize for 10–20 min at 30 °C in an Eppendorf tube. Subsequently, the sample was gently resuspended and 4 µl was transferred to a glow-discharged holey carbon R2/1 copper grid (Quantifoil Micro Tools) suspended in the chamber of a Vitrobot (Thermo Fisher Scientific). The sample was incubated for 2 min inside the Vitrobot chamber, equilibrated at 30 °C and 95% relative humidity, to allow for potential MT repolymerization after sample transfer to the grid. Subsequently, grids were manually back-blotted for 3 s and plunge-frozen in liquid ethane. Vitrified grids were clipped into autogrid cartridges and stored in liquid nitrogen until future use.

**Single-molecule fluorescence intensity analysis**
Protein samples of GFP, GFP–EB3 and GFP–CEP104, GFP–TOGARAM1 or GFP–ARMC9 were diluted in PBS and immobilized in adjacent flow chambers of the sample plasma-cleaned glass coverslip as described previously[68]. Flow chambers were then sealed with vacuum grease and immediately imaged using TIRF microscopy. A total of 10–20 images (per condition) of previously unexposed coverslip areas were acquired with a 200-ms exposure time.

**CEP104 molecule counting at blocked MT plus ends**
To determine the number of CEP104 molecules at blocked MT plus ends, we immobilized GFP in one chamber of the same plasma-cleaned glass coverslip and in vitro reconstitution assay in the adjacent chamber (as described above). Chambers were sealed and immediately imaged using TIRF microscopy. First images on unbleached GFP single molecules were acquired; then, using the same illumination conditions, images of unexposed GFP–CEP104 bound MTs were acquired.

**FRAP**
For FRAP experiments, in vitro reconstitutions were prepared as described above and imaging was conducted using TIRF microscopy. A focused 488-nm laser (for GFP–EB3) or 561-nm laser (for mCherry–CEP104) was used to bleach specific regions of the MT lattice.

**Microscopy**

**TIRF microscopy.** In vitro reconstitution assays imaged on a previously described (iLas2) TIRF microscope setup[34]. We used the iLas3 system (Gataca Systems), which is a dual laser illuminator for azimuthal spinning TIRF (or Hilo) illumination. This system was installed on Nikon Ti microscope (with the perfect focus system, Nikon), equipped with a 489-nm 150-mW Vortran Stradus laser (Vortran), 100-mW 561-nm OBIS laser (Coherent) and 49002 and 49008 Chroma filter sets. Additionally, a charge-coupled device (CCD) camera CoolSNAP MYO (Teledyne Photometrics) was installed and the set up was controlled with MetaMorph 7.10.2.240 software (Molecular Device). To keep the in vitro samples at 30 °C, a stage-top incubator model INUBG2E-ZILCS (Tokai Hit) was used. The final resolution using the CCD camera was 0.045 µm per pixel. For all assays, MT dynamics was measured at 200-ms exposure and 3-s intervals for 10 min. For EB3 FRAP experiments continuous imaging was used with 100-ms exposure. For kinesin experiments, a 10-min video was first acquired at 200-ms exposure and 3-s intervals to measure MT dynamics; subsequently, DmKHC (1–421)–SNAP–6xHis (gift from Kapitein Labs; Addgene, 196976) labeled with Alexa647–SNAP dye (New England Biolabs) was continuously imaged using 100-ms exposure.

**Cryo-ET.** Vitrified in vitro reconstituted MTs were imaged on a Talos Arctica transmission electron microscope (200 keV) (Thermo Fisher Scientific), equipped with a K2 summit direct electron detector (Gatan) and postcolumn energy filter aligned to the zero-loss peak with 20-keV slit width (controlled through DigitalMicrograph 3.32). Image acquisition was controlled by Serial-EM 4.1 beta11 (ref. 69). MT ends were located on transmission electron microscopy overview images, acquired at ×4,100 magnification (33.3 Å per pixel). Tilt series were collected at ×63,000 magnification (2.17 Å per pixel) with a dose rate of ~5 e⁻ per pixel per s and a total dosage of 100 e⁻ per Å². Tilt images were recorded using a dose-symmetric scheme[70] over a tilt range of 54° to −54° or 60° to −60° with a tilt increment of 2°, at a defocus target of −3 µm.

**Image analysis**

**Analysis of MT plus-end dynamics in vitro.** Analysis of MT plus-end dynamics was performed as described before[34]. Videos of dynamic MTs were corrected for drift and kymographs were generated using the plugin KymoResliceWide version 0.4 in ImageJ (https://github.com/ekatrukha/KymoResliceWide). MT tips were traced with lines and measured lengths and angles were used to calculate the MT dynamic parameters such as growth rate and transition events. All events with growth rates faster than 0.02 µm min⁻¹ and slower than 0.5 µm min⁻¹ were categorized as slow growing, events faster than 0.5 µm min⁻¹ were categorized as fast growing and events with shrinkage rates faster than 0.02 µm min⁻¹ were categorized as shrinking. The events with slower growth rates or faster shrinkage rates than the aforementioned rates were categorized as pause events. Transition frequencies were calculated by dividing the sum of the transition events per experiment by the total time this event could have occurred. Weighted growth rate histograms were calculated by taking individual growth durations and dividing by total growth time.

**Analysis of FRAP experiments.** Data were normalized to 1 for image acquired immediately before the FRAP event and 0 for the image acquired immediately after the FRAP event using the following equation: normalized intensity = (intensity$_{t=x}$ − intensity$_{t=1}$)(intensity $_{t=0}$ − intensity$_{t=1}$). For EB3, nonlinear fit curves were fitted to the postbleach intensities and the half-life was calculated using GraphPad Prism.

**Analysis of single-molecule fluorescence intensities.** Single-molecule fluorescence spots of proteins immobilized directly on the cover glass were detected and fitted with a 2D Gaussain function

using custom-written ImageJ plugin DoM-Utrect (https://github.com/ekatrukha/DoM_Utrecht). Fitted peak intensities were then used to create intensity histograms. CEP104-blocked plus ends were then manually annotated and fitted with a 2D Gaussian function using the same plugin.

**Cryo-ET 3D volume reconstruction and analysis.** Tomogram reconstruction and denoising were performed using an in-house Python script, as described previously[71]. The tilt series' frames were initially aligned and dose-weighted using MotionCor2 (ref. 72). Subsequent tilt series alignment and tomographic reconstruction for denoising, visualization and analysis were performed through AreTomo 1.3.3. Final reconstructed tomographic volumes were created through weighted backprojection and binned with a factor of 2. Cryo-CARE 0.2.2 and 0.3.0 were used for tomogram denoising[73]. For this, even and odd tomographic reconstructions were generated through splitting of the video frames for each individual tilt into even and odd summed frames with MotionCor2 (ref. 72). Alignment parameters were first calculated on the full tilt series and subsequently applied to the even and odd stacks. The cryo-CARE network was trained on five tomograms per dataset and then applied to the entire set.

For visualization purposes, semiautomated segmentation was performed on denoised tomograms using the EMAN2 2.91 and 2.99 tomoseg module[74]. For this, separate neural networks were trained for the two features of interest: 'MT' and 'CTM'. Each feature was included as 'negative' training reference, together with background noise and ice contamination, for the neural network training of the other feature. Final visualization and 3D rendering was performed in ChimeraX.

MT ends, both free and plugged with CTM proteins, were only analyzed for protofilament shape in case they were plus ends. MT polarity was determined through 2D rotational averaging with applied symmetry, using the EMAN2 program e2proc2d. Manual 3D tracing of protofilament shapes at MT plus ends was performed as previously described[51,63,75] in the IMOD 4.9.6 program 3dmod (ref. 76). Manual tracing was performed on tomographic subvolumes, generated with the mtrotlong program of IMOD (4.11.20). Although the missing wedge stretching complicated resolving neighboring protofilaments, most protofilaments could be distinguished with manual tracing because of differences in length or their deviation from the plane parallel to the imaging direction (because of flaring). Additionally, the total protofilament number of MT tips was determined before tracing using 2D rotational symmetry averaging of the MT cross-section. We looked for neighboring protofilaments at angular distances corresponding to the determined protofilament number (360°/protofilament number) with an error margin of ±4° to account for potentially imperfect tilt alignments of tomograms. Furthermore, to faithfully trace protofilaments at specific orientations, the tracing procedure was monitored from orthogonal views using 3dmod zap and slicer windows, as well as the volume viewer to show the electron density signal in 3D. Nonetheless, because of the signal-to-noise ratio in the tomograms, we cannot exclude the possibility of slight inaccuracies in the manual tracing, especially with extra densities present from the CTM. Protofilaments that could not be confidently traced were excluded from the analysis.

Traces of individual protofilaments were saved as separate contours within one object per MT. To satisfy the three-point requirement of howflared, for blunt end protofilaments, a third point was added one pixel upstream of the second point that indicated the last protofilament segment in the MT wall. Subsequently, protofilament coordinates were extracted using the howflared program of IMOD (4.11.20). MT raggedness and protofilament length parameters, as described in the Results, were calculated by howflared. Curvature analysis was performed on the howflared output with Matlab scripts available from GitHub (https://github.com/ngudimchuk/Process-PFs). The total number of analyzed MTs and protofilaments can be found in the legend of Fig. 8. Protofilament tracing and analysis were performed on two biological replicate datasets for the control condition (dataset 1, $n$ = 8 MTs; dataset 2, $n$ = 18 MTs) and three biological replicate datasets for the CTM condition (dataset 1, $n$ = 20 MTs; dataset 2, $n$ = 6 MTs; dataset 3, $n$ = 7 MTs).

**Statistical analysis**
Statistical analysis was performed using GraphPad Prism version 9. Figure legends contain statistical details of each experiment, including the statistical tests used, the number of measurements and the number of experiments.

**Reporting summary**
Further information on research design is available in the Nature Portfolio Reporting Summary linked to this article.

## Data availability
MS data generated for this paper were deposited to the ProteomeXchange Consortium through the PRIDE partner repository with the dataset identifier PXD055631. Cryo-ET data generated for the paper were deposited to the Electron Microscopy Data Bank (EMDB) and Electron Microscopy Public Image Archive (EMPIAR). All tomographic subvolumes on which MT protofilament tracing was performed, as well as representative full volume tomograms, are available from the EMDB through the following accession codes: EMD-52359 for the control condition (MTs grown with EB3 alone) and EMD-52360 for the CTM condition (MTs grown with the CTM and EB3). The raw tilt series data and the protofilament tracing models are available from EMPIAR through the following accession codes: EMPIAR-12498 for the CTM condition (MTs grown with the CTM and EB3) and EMPIAR-12499 for the control condition (MTs grown with EB3 alone). All other data that support conclusions of this paper are available in the manuscript itself or from the authors on request. All unique and stable reagents generated in this study are available from the lead contact without restriction. Source data are provided with this paper.

## Code availability
The python script for automated tomogram reconstruction and cryo-CARE denoising is available from GitHub (https://github.com/SBC-Utrecht/cryocare-from-movies). MATLAB scripts for analysis of protofilament tracings in IMOD are available from GitHub (https://github.com/ngudimchuk/Process-PFs).

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

## Acknowledgements

We thank members of the A.A. and S.C.H. labs for insightful discussions. We also thank V. Volkov (Queen Mary, University of London) for advice on cryo-ET sample preparation and V. Volkov and N. Gudimchuk (Lomonosov Moscow State University) for help with cryo-ET analysis. Electron imaging was performed at the Utrecht University Electron Microscopy Center. We thank I. C. Schneijdenberg and M. Bergmeijer for cryo-ET microscopy support. This work was supported by the European Research Council (synergy grant 609822 to A.A.), the Netherlands Organization for Health Research and Development (ZonMw) (grant 09120012110085 to R.R. and A.A.), Dutch Research Council (NWO) (grant OCENW.XL21.XL21.048 to A.A. and S.C.H.) and European Molecular Biology Organization long-term fellowship (AFTL 74-2022 to H.A.J.S.).

## Author contributions

H.A.J.S. and C.M.v.d.B. performed the experiments, analyzed the data and wrote the paper. R.A.H. performed and analyzed the cryo-ET experiments and wrote the paper. D.S. performed the immunoprecipitation experiments. K.E.S. performed and analyzed the MS experiments. S.H. supervised and analyzed the cryo-ET experiments. R.R generated essential reagents and acquired funding. A.A. coordinated the project, acquired funding and wrote the paper. All authors reviewed and edited the paper.

## Competing interests

The authors declare no competing interests.

## Additional information

**Extended data** is available for this paper at https://doi.org/10.1038/s41594-025-01483-y.

**Correspondence and requests for materials** should be addressed to Anna Akhmanova.

**a**

SII-GFP-ARMC9 FL, SII-GFP-CCDC66 FL, SII-GFP-CEP104 FL, SII-GFP-CSPP1 FL, SII-GFP-TOGARAM1 FL

**d**

mCH-CEP104 (10 nM), HiLyte488-tubulin

**e**

mCH-CEP104 (100 nM), HiLyte488-tubulin

DmKHC(1–421)

**b**

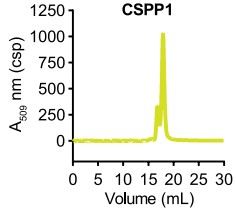

**c**

| Accession | Description | Coverage (%) | PSMs (#) | Abundance | Normalized abundance (%) |
|---|---|---|---|---|---|
| **Q723E5** | **LisH domain-containing protein ARMC9 [ARMC9]** | **65** | **4353** | **1.53E+11** | **100.00** |
| P68363 | Tubulin alpha-1B chain [TUBA1B] | 31 | 11 | 2.85E+07 | 0.02 |
| P07437 | Tubulin beta chain [TUBB] | 18 | 7 | 1.23E+07 | 0.01 |
| P60709 | Actin, cytoplasmic 1 [ACTB] | 11 | 4 | 5.89E+06 | 0.00 |

| Accession | Description | Coverage (%) | PSMs (#) | Abundance | Normalized abundance (%) |
|---|---|---|---|---|---|
| **A2RUB6** | **Coiled-coil domain-containing protein 66 [CCDC66]** | **84** | **2750** | **6.50E+10** | **100.00** |
| A0A0G2JIW1 | Heat shock 70 kDa protein 1B [HSPA1B] | 89 | 880 | 1.85E+10 | 39.00 |
| P07437 | Tubulin beta chain [TUBB] | 50 | 27 | 5.53E+07 | 0.03 |
| Q1MSJ5 | Centrosome and spindle pole-associated protein 1 [CSPP1] | 29 | 25 | 2.22E+07 | 0.08 |
| P68371 | Tubulin beta-4B chain [TUBB4B] | 40 | 25 | 2.17E+06 | 0.00 |
| F5H5D3 | Tubulin alpha chain [TUBA1C] | 35 | 21 | 3.27E+06 | 0.06 |
| P68363 | Tubulin alpha-1B chain [TUBA1B] | 46 | 21 | 7.55E+07 | 0.01 |

| Accession | Description | Coverage (%) | PSMs (#) | Abundance | Normalized abundance (%) |
|---|---|---|---|---|---|
| **O60308** | **Centrosomal protein of 104 kDa [CEP104]** | **88** | **6216** | **1.49E+11** | **100.00** |
| Q1MSJ5 | Centrosome and spindle pole-associated protein 1 [CSPP1] | 43 | 50 | 5.36E+08 | 0.36 |
| Q14008 | Cytoskeleton-associated protein 5 [CKAP5] | 11 | 38 | 4.50E+07 | 0.03 |
| P07437 | Tubulin beta chain [TUBB] | 57 | 30 | 1.25E+08 | 0.08 |
| P68371 | Tubulin beta-4B chain [TUBB4B] | 51 | 25 | 6.64E+06 | 0.00 |
| Q71U36 | Tubulin alpha-1A chain [TUBA1A] | 42 | 22 | 8.53E+07 | 0.06 |
| Q8IW35 | Centrosomal protein of 97 kDa [CEP97] | 19 | 13 | 1.33E+07 | 0.01 |
| Q7Z3E5 | LisH domain-containing protein ARMC9 [ARMC9] | 7 | 3 | 1.54E+06 | 0.00 |
| A6NHL2 | Tubulin alpha chain-like 3 [TUBAL3] | 6 | 2 | 1.57E+06 | 0.00 |

| Accession | Description | Coverage (%) | PSMs (#) | Abundance | Normalized abundance (%) |
|---|---|---|---|---|---|
| **Q1MSJ5** | **Centrosome and spindle pole-associated protein 1 [CSPP1]** | **84** | **3785** | **1.01E+11** | **100.00** |
| A0A0G2JIW1 | Heat shock 70 kDa protein 1B [HSPA1B] | 89 | 747 | 2.18E+10 | 21.63 |
| A0A5H1ZRS1 | Pericentriolar material 1 protein [PCM1] | 16 | 25 | 2.46E+07 | 0.02 |
| P07437 | Tubulin beta chain [TUBB] | 49 | 24 | 7.88E+07 | 0.08 |
| P68371 | Tubulin beta-4B chain [TUBB4B] | 46 | 21 | 3.94E+06 | 0.00 |
| Q71U36 | Tubulin alpha-1A chain [TUBA1A] | 53 | 21 | 1.19E+08 | 0.12 |
| F5H5D3 | Tubulin alpha chain [TUBA1C] | 36 | 19 | 1.09E+07 | 0.01 |
| P04350 | Tubulin beta-4A chain [TUBB4A] | 40 | 17 | 1.42E+06 | 0.00 |

| Accession | Description | Coverage (%) | PSMs (#) | Abundance | Normalized abundance (%) |
|---|---|---|---|---|---|
| **Q6A070** | **TOG array regulator of axonemal microtubules protein 1 [TOGARAM1]** | **60** | **2592** | **3.37E+10** | **100.00** |
| P07437 | Tubulin beta chain [TUBB] | 40 | 56 | 4.34E+07 | 0.13 |
| P68371 | Tubulin beta-4B chain [TUBB4B] | 44 | 49 | 1.07E+08 | 0.32 |
| Q13885 | Tubulin beta-2A chain [TUBB2A] | 38 | 43 | 1.31E+06 | 0.00 |
| Q71U36 | Tubulin alpha-4B chain [TUBA1A] | 45 | 33 | 1.77E+06 | 0.01 |
| P68363 | Tubulin alpha-1B chain [TUBA1B] | 45 | 33 | 1.67E+08 | 0.49 |
| Q7Z3E5 | LisH domain-containing protein ARMC9 [ARMC9] | 14 | 8 | 6.92E+08 | 0.02 |
| P60709 | Actin, cytoplasmic 1 [ACTB] | 25 | 8 | 1.30E+07 | 0.01 |
| O60308 | Centrosomal protein of 104 kDa [CEP104] | 7 | 4 | 2.31E+06 | 0.01 |
| A6NHL2 | Tubulin alpha chain-like 3 [TUBAL3] | 6 | 3 | 2.64E+06 | 0.01 |

**Extended Data Fig. 1 | See next page for caption.**

**Extended Data Fig. 1 | Characterization of CTM proteins in vitro. a**, Analysis of purified GFP-tagged CTM proteins by SDS-PAGE. Asterisks indicate the full-length protein bands. Protein concentrations were determined from a BSA standard. Assays were repeated independently at least two times. **b**, SEC-MALS analysis of purified GFP-tagged CTM proteins. The human sequence was used for all module members, except for TOGARAM1. Due to technical difficulties, the mouse sequence of TOGARAM1 was used, which is 84% identical to human TOGARAM1. **c**, Mass spectrometry analysis of purified GFP-tagged CTM proteins. **d**, Fields of view (top, scale bar 2 μm) and kymograph (bottom, scale bars 2 μm and 60 s) illustrating MT dynamics from GMPCPP-stabilized seed with HiLyte-488-tubulin and mCherry-CEP104. **e**, Kymographs illustrating mobility of DmKHC(1-421) on CEP104-blocked MT labelled with HiLyte-488-tubulin and mCherry-CEP104 (top) and DmKHC (bottom) proving that the blocked end of the MT is the plus end. Scale bars 2 μm and 60 s for both kymographs.

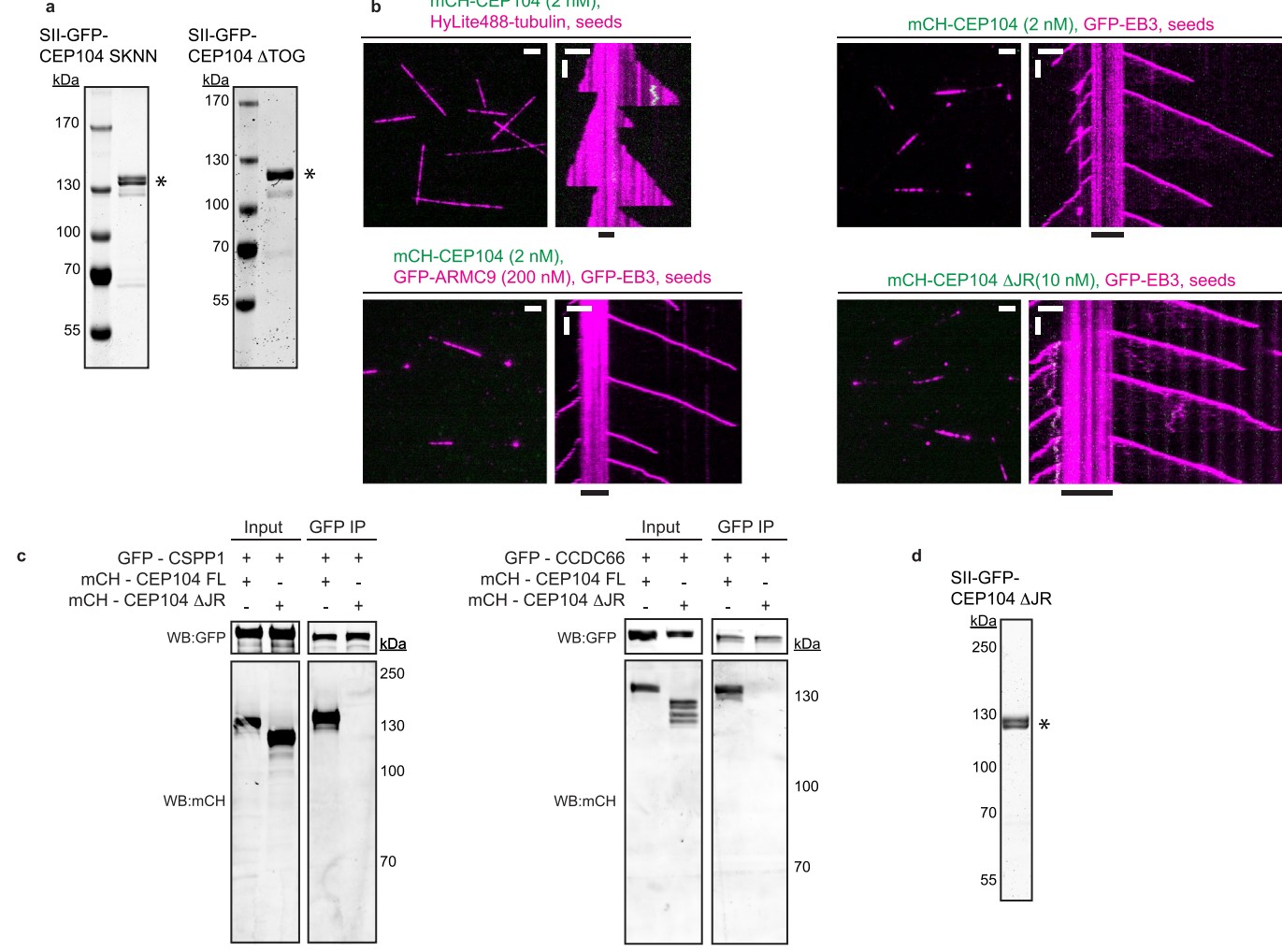

**Extended Data Fig. 2 | Characterization of CEP104 constructs in vitro.**
**a**, Analysis of purified GFP-tagged CEP104 constructs by SDS-PAGE. Asterisks show protein bands. Protein concentrations were determined from a BSA standard. Assays were repeated independently at least two times. **b**, Fields of view (left, scale bar 2 µm) and kymograph (right, scale bars 2 µm and 60 s) illustrating MT dynamics from GMPCPP-stabilized seed with either HiLyte-488-tubulin or GFP-EB3 and indicated concentrations and colors of CEP104 constructs.

**c**, Co-immunoprecipitation of either CSPP1 (left) or CCDC66 (right) with indicated CEP104 constructs, both CSPP1 and CCDC66 interact with the jelly-roll domain of CEP104. Assays were repeated independently at least two times.
**d**, Analysis of purified GFP-tagged CEP104 ΔJR construct by SDS-PAGE. Asterisk indicates the full-length protein band. Protein concentration was determined from a BSA standard. Assays were repeated independently at least two times.

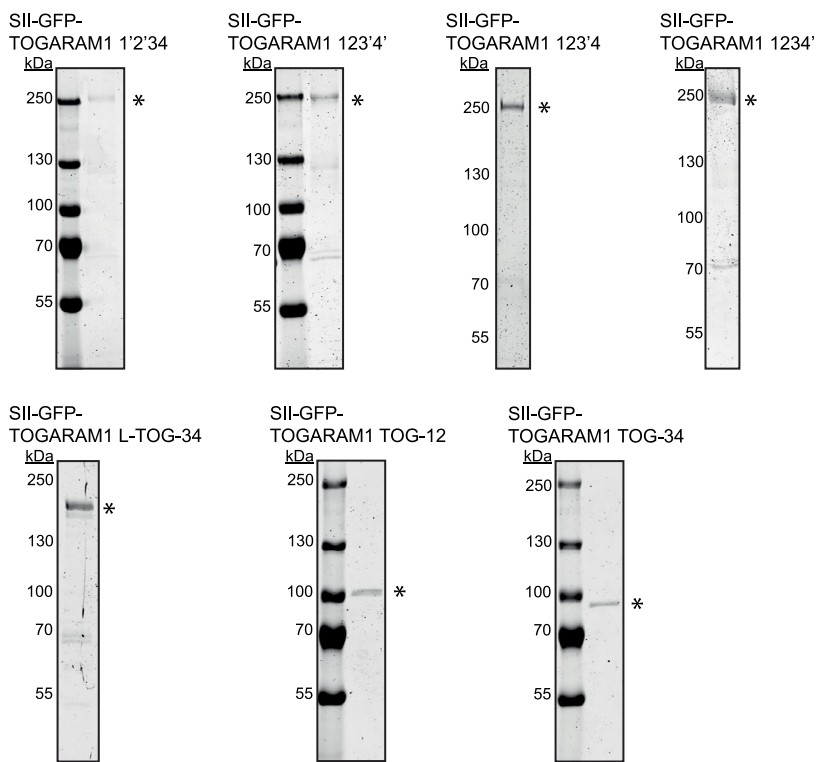

**Extended Data Fig. 3 | Characterization of TOGARAM1 constructs in vitro.** Analysis of purified GFP-tagged TOGARAM1 constructs by SDS-PAGE. Asterisks show protein bands. Protein concentrations were determined from a BSA standard. Assays were repeated independently at least two times.

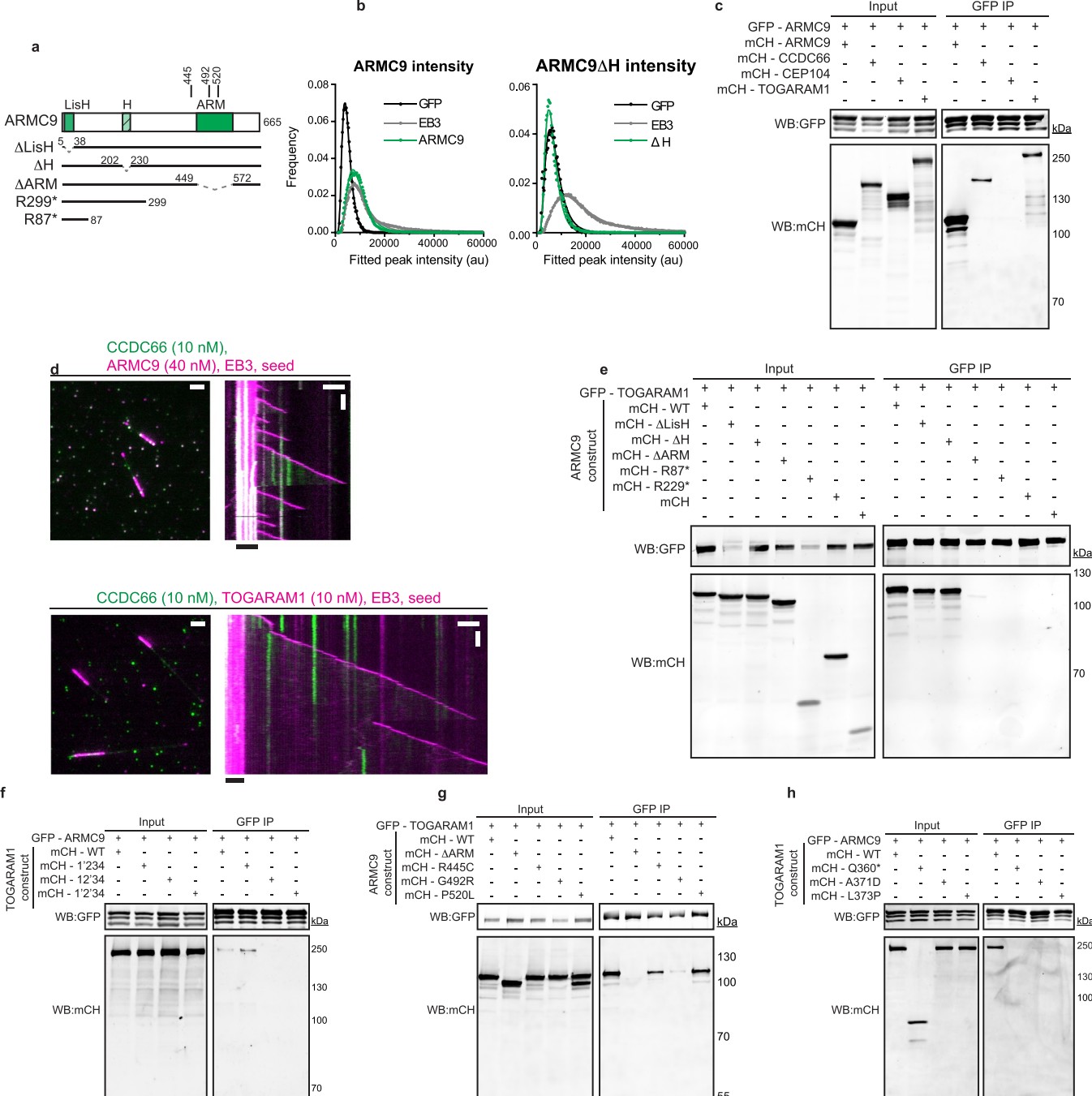

**Extended Data Fig. 4 | Mapping of ARMC9 and TOGARAM1 interaction.**
**a**, Schematic representation of different ARMC9 constructs. Vertical lines indicate Joubert syndrome-linked point mutations. **b**, Histogram plots of fluorescent intensities of single GFP molecules, GFP-EB3 dimers, and full-length GFP-ARMC dimers (left) or GFP-ARMC9 ΔH molecules immobilized in separate chambers of the same coverslips. Number of molecules analyzed, left, right: GFP, n=18415,14914; GFP-EB3, n=132826,163959; GFP-ARMC9 FL, n=99873; GFP-ARMC9 ΔH, 55479. **c**, Co-immunoprecipitation of full-length ARMC9 with indicated proteins. Assays were repeated independently at least two times. **d**, Fields of view (left, scale bar 2 μm) and kymograph (right, scale bars 2 μm and

60 s) illustrating MT dynamics from GMPCPP-stabilized seed with mCherry-EB3 and indicated concentrations and colors of CTM proteins. Assays were repeated independently three times. **e, f**, Co-immunoprecipitation experiments of either full-length TOGARAM1 with indicated ARMC9 constructs (**e**) or full-length ARMC9 with indicated TOGARAM1 point mutations (**f**). ARMC9 and TOGARAM1 interact through ARM domain and TOG2 domain, respectively. Assays were repeated independently at least two times. **g, h**, Co-immunoprecipitation experiments of either full-length TOGARAM1 with indicated ARMC9 Joubert syndrome mutations (**g**) or full-length ARMC9 with indicated TOGARAM1 Joubert syndrome mutations (**h**). Assays were repeated independently at least two times.

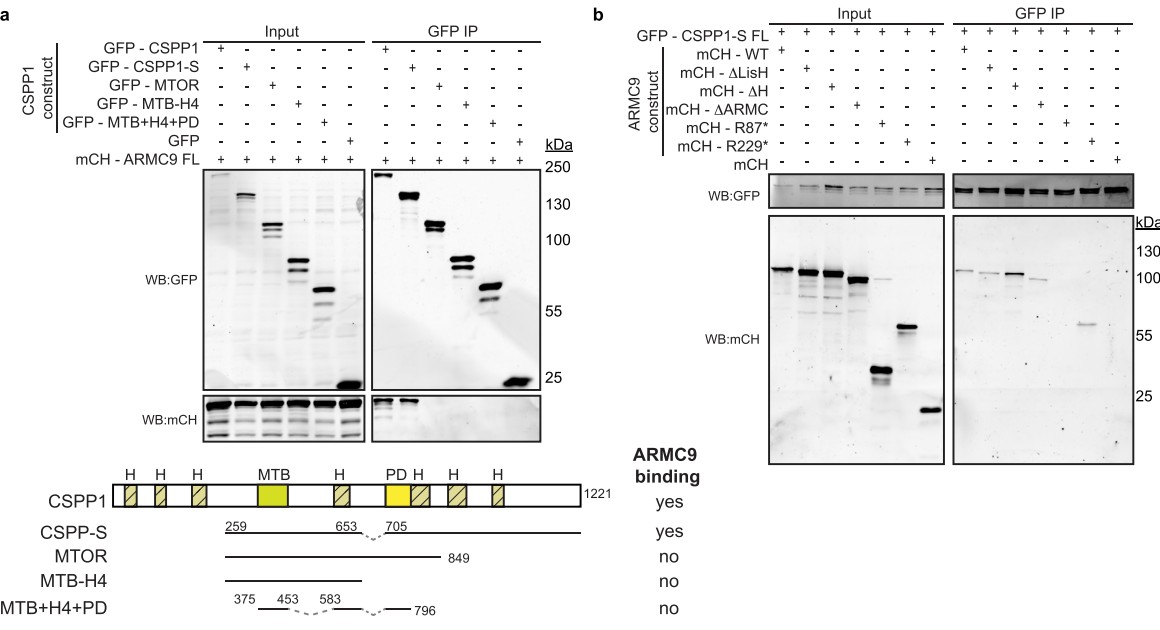

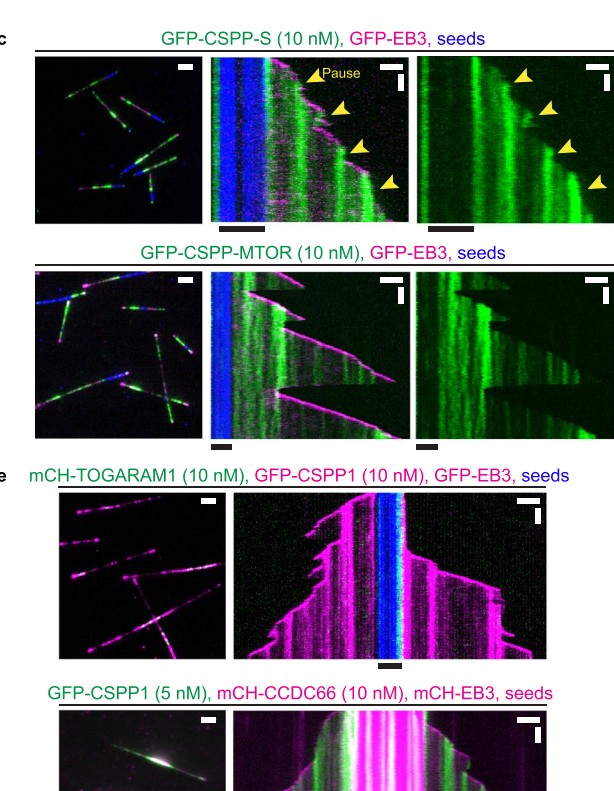

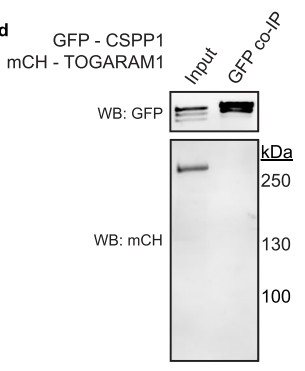

**Extended Data Fig. 5 | Mapping of ARMC9 and CSPP1 interaction.**
**a**, Co-immunoprecipitation experiment of full-length ARMC9 with indicated CSPP1 constructs illustrated in schematic below. Assays were repeated independently at least two times. **b**, Co-immunoprecipitation experiment of full-length CSPP1 with indicated ARMC9 constructs illustrated in Extended Data Fig. 4a. Assays were repeated independently at least two times. **c**, Fields of view (left, scale bar 2 μm) and kymographs (right, scale bars 2 μm and 60 s) illustrating MT dynamics from GMPCPP-stabilized seed with GFP-EB3 and

indicated concentrations and colors of CSPP1 constructs. Assays were repeated independently three times. **d**, Co-immunoprecipitation experiment of full-length CSPP1 with full-length TOGARAM1 shows no interaction. Assays were repeated independently at least two times. **e**, Fields of view (left, scale bar 2 μm) and kymograph (right, scale bars 2 μm and 60 s) illustrating MT dynamics from GMPCPP-stabilized seed with GFP- or mCherry-EB3 and indicated concentrations and colors of CTM proteins. Assays were repeated independently three times.

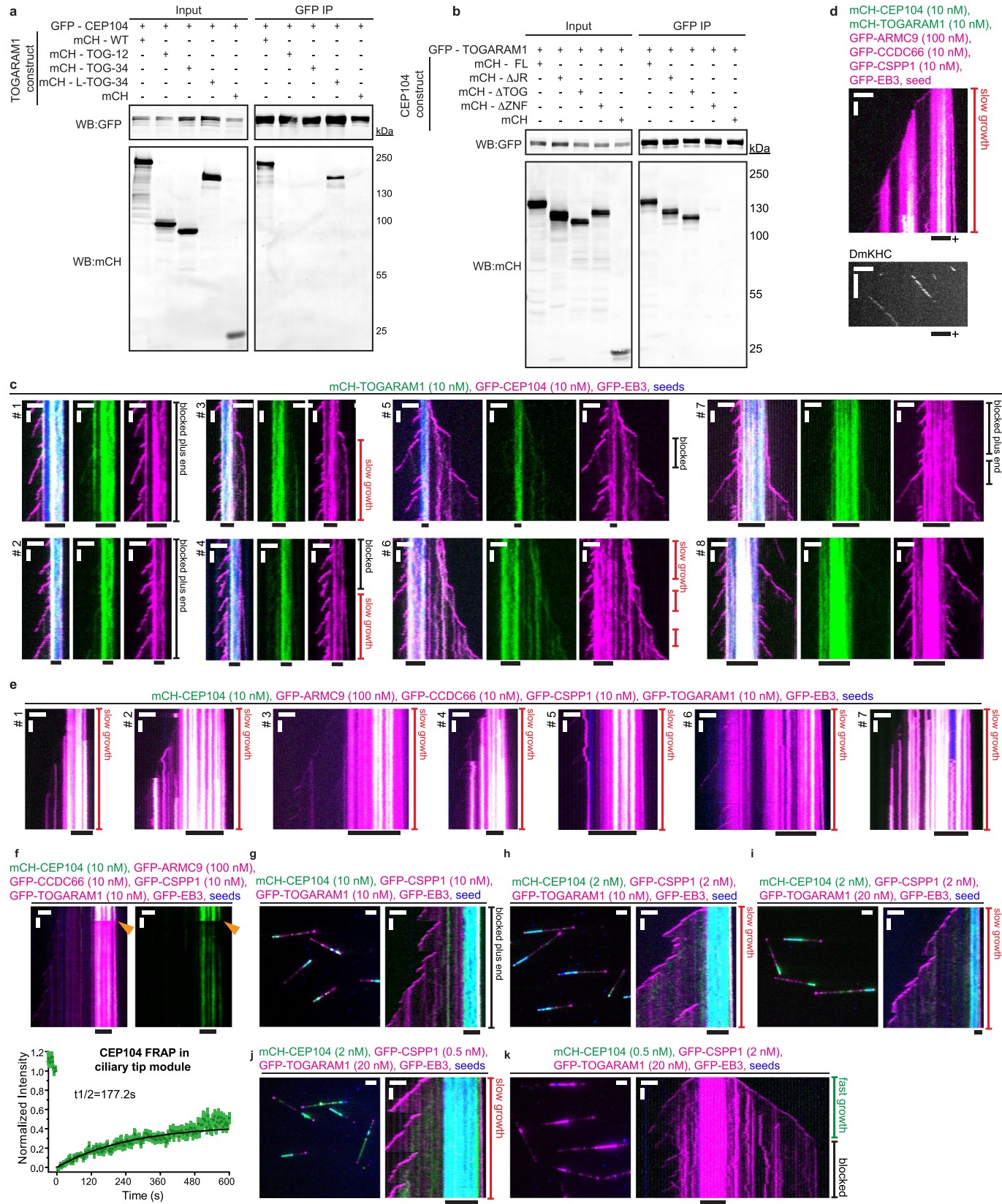

**Extended Data Fig. 6 | See next page for caption.**

**Extended Data Fig. 6 | Mapping of CEP104 and TOGARAM1 interaction.**
**a, b**, Co-immunoprecipitation of either wildtype CEP104 with indicated
TOGARAM1 constructs (**a**) or wildtype TOGARAM1 with indicated CEP104
constructs (**b**), the two proteins interact between the linker of TOGARAM1
and the zinc finger of CEP104. Assays were repeated independently at least
two times. **c**, Additional kymographs (scale bars 2 μm and 60 s) illustrating MT
dynamics from GMPCPP-stabilized seed with GFP-EB3, mCherry-TOGARAM1,
and GFP-CEP104. Assay was repeated independently three times. **d**, Kymographs
illustrating mobility of DmKHC(1-421) on slow growing MT with all CTM proteins
(top) and DmKHC (bottom) proving that the slow growing end of the MT is the
plus end. Scale bars 2 μm and 60 s for both kymographs. Assay was repeated
independently three times **e**, Additional kymographs (scale bars 2 μm and 60 s)
illustrating MT dynamics from GMPCPP-stabilized seeds of the entire CTM
with 20 nM GFP-EB3. Assay was repeated independently three times.
**f**, FRAP analysis of CEP104 at slow growing MT plus ends with the entire CTM.
Arrowhead marks point of photobleaching in representative kymograph (top),
scale bars 2 μm and 60 s. Plot (bottom) show average curve with exponential
fit. Number of FRAP measurements, n=19 Error bars represent s.e.m. Assay was
repeated independently three times. **g-k**, Fields of view (left, scale bar 2 μm)
and kymographs (right, scale bars 2 μm and 60 s) illustrating MT dynamics from
GMPCPP-stabilized seeds with 20 nM GFP-EB3 and indicated concentrations and
colors of CTM proteins. Assays were repeated independently three times.

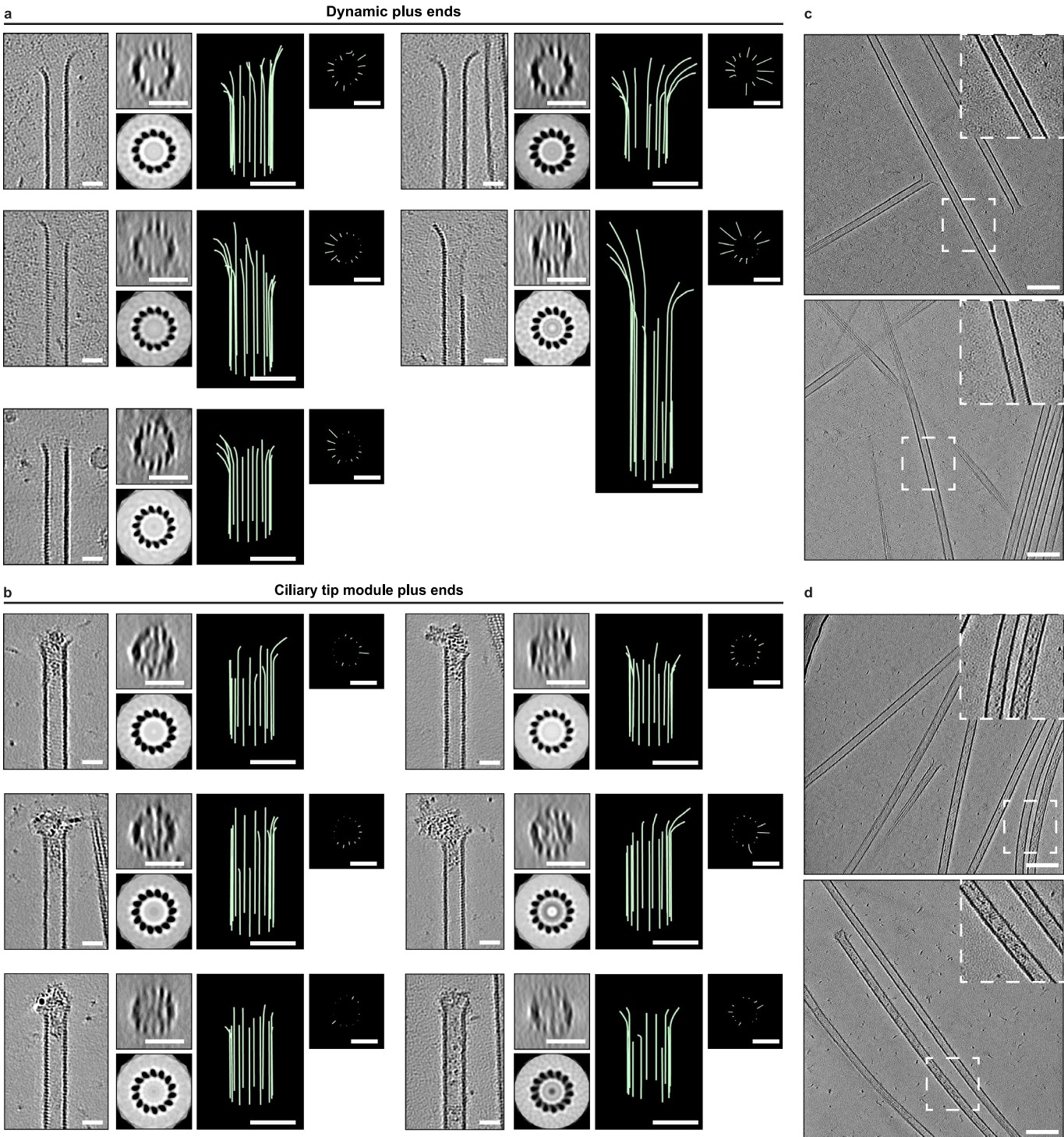

**Extended Data Fig. 7 | Characterization of MT plus ends by cryo-ET.**
**a, b**, Representative examples of MT plus ends grown in presence of EB3 alone (**a**) or CTM and EB3 (**b**). Per MT, the following is shown from left to right; a slice of the denoised tomogram containing the MT plus end, an 8 nm thick transverse cross-section accompanied by rotational averaging analysis to determine MT polarity, and the corresponding 3D model of manual protofilament tracing accompanied by its transverse cross-section. Scale bars 25 nm. Independent experiments were run twice for controls and three time for CTM samples. **c, d**, Slices (4.3 nm thick) of denoised tomograms showing MTs grown in presence of EB3 alone (**c**) or the CTM and EB3 (**d**). Insets show a zoom of the indicated region. Scale bars 100 nm.

# Reporting Summary

## Statistics

For all statistical analyses, confirm that the following items are present in the figure legend, table legend, main text, or Methods section.

| n/a | Confirmed | |
|---|---|---|
| ☐ | ☒ | The exact sample size (*n*) for each experimental group/condition, given as a discrete number and unit of measurement |
| ☐ | ☒ | A statement on whether measurements were taken from distinct samples or whether the same sample was measured repeatedly |
| ☐ | ☒ | The statistical test(s) used AND whether they are one- or two-sided *Only common tests should be described solely by name; describe more complex techniques in the Methods section.* |
| ☒ | ☐ | A description of all covariates tested |
| ☐ | ☒ | A description of any assumptions or corrections, such as tests of normality and adjustment for multiple comparisons |
| ☐ | ☒ | A full description of the statistical parameters including central tendency (e.g. means) or other basic estimates (e.g. regression coefficient) AND variation (e.g. standard deviation) or associated estimates of uncertainty (e.g. confidence intervals) |
| ☐ | ☒ | For null hypothesis testing, the test statistic (e.g. *F*, *t*, *r*) with confidence intervals, effect sizes, degrees of freedom and *P* value noted *Give P values as exact values whenever suitable.* |
| ☒ | ☐ | For Bayesian analysis, information on the choice of priors and Markov chain Monte Carlo settings |
| ☒ | ☐ | For hierarchical and complex designs, identification of the appropriate level for tests and full reporting of outcomes |
| ☒ | ☐ | Estimates of effect sizes (e.g. Cohen's *d*, Pearson's *r*), indicating how they were calculated |

*Our web collection on statistics for biologists contains articles on many of the points above.*

## Software and code

Policy information about availability of computer code

| Data collection | MetaMorph 7.10.2.240 software, Digital micrograph 3.32, Serial-EM 4.1 beta11 |
|---|---|
| Data analysis | ImageJ 1.45s, ImageJ 1.53c (Fiji), ImageJ plugin KymoResliceWide v.0.4 (https://github.com/ekatrukha/KymoResliceWide), ImageJ plugin DoM_Utrecht v.1.1.6 (https://github.com/ekatrukha/DoM_Utrecht), AreTomo 1.3.3 , Cryo-CARE 0.2.2, Cryo-CARE 0.3.0, MotionCor2, EMAN2 2.91, EMAN2 2.99, IMOD 4.9.6, IMOD 4.11.20, Matlab R2022b, Matlab R2024a, Matlab scripts for g custom algorithms or softwarcurvature analysis (https://github.com/ngudimchuk/Process-PFs) that are central to the research. Python scripts for automated tomogram reconstruction and cryo-CARE denoising (https://github.com/SBC-Utrecht/cryocare-from-movies). |

For manuscripts utilizing custom algorithms or software that are central to the research but not yet described in published literature, software must be made available to editors and reviewers. We strongly encourage code deposition in a community repository (e.g. GitHub). See the Nature Portfolio guidelines for submitting code & software for further information.

## Data

Policy information about availability of data

All manuscripts must include a data availability statement. This statement should provide the following information, where applicable:
- Accession codes, unique identifiers, or web links for publicly available datasets
- A description of any restrictions on data availability
- For clinical datasets or third party data, please ensure that the statement adheres to our policy

Mass spectrometry data has been deposited to the ProteomeXchange Consortium via the PRIDE partner repository with the dataset identifier PXD055631. Cryo electron tomography data generated for the paper have been deposited to EMDB with the accession codes EMD-52359 and EMD-52360, and to EMPIAR with the accession codes EMPIAR-12498 and EMPIAR-12499. The EMDB and EMPIAR depositions will become publicly available once the paper is released. All data that support the conclusions are either available in the manuscript itself or available from the authors on request.

## Research involving human participants, their data, or biological material

Policy information about studies with human participants or human data. See also policy information about sex, gender (identity/presentation), and sexual orientation and race, ethnicity and racism.

| | |
|---|---|
| Reporting on sex and gender | n/a |
| Reporting on race, ethnicity, or other socially relevant groupings | n/a |
| Population characteristics | n/a |
| Recruitment | n/a |
| Ethics oversight | n/a |

Note that full information on the approval of the study protocol must also be provided in the manuscript.

# Field-specific reporting

Please select the one below that is the best fit for your research. If you are not sure, read the appropriate sections before making your selection.

☒ Life sciences   ☐ Behavioural & social sciences   ☐ Ecological, evolutionary & environmental sciences

For a reference copy of the document with all sections, see nature.com/documents/nr-reporting-summary-flat.pdf

# Life sciences study design

All studies must disclose on these points even when the disclosure is negative.

| | |
|---|---|
| Sample size | No statistical methods were used to predict the sample size. All datasets were pooled from at least three or more independent experiments. Sample size was chosen based on the reproducibility, our previous experience or the standards in the field. Data distribution was assumed to be normal but this was not formally tested. In each case, sample size, number of independent experiments and statistical tests, when used, along with p values were indicated in the figure panels, legends or within the methods section. |
| Data exclusions | No data were excluded from the analyses. |
| Replication | Each experimental condition was repeated at least three times or more unless stated otherwise. All attempts at replication were successful. |
| Randomization | No randomization was performed in our study as samples were not required to be allocated into experimental groups. |
| Blinding | Investigators were not blinded to group allocation during data collection and analyses as group allocation was not required. Data was collected immediately after each experiment was performed for each experimental condition with different proteins or different protein concentrations, and therefore blinding was not possible. The analyses were done using automated methods or using strictly defined parameters as mentioned in the methods section, therefore preventing any subjective error. |

# Reporting for specific materials, systems and methods

We require information from authors about some types of materials, experimental systems and methods used in many studies. Here, indicate whether each material, system or method listed is relevant to your study. If you are not sure if a list item applies to your research, read the appropriate section before selecting a response.

## Materials & experimental systems

| n/a | Involved in the study |
|---|---|
| ☐ | ☒ Antibodies |
| ☐ | ☒ Eukaryotic cell lines |
| ☒ | ☐ Palaeontology and archaeology |
| ☒ | ☐ Animals and other organisms |
| ☒ | ☐ Clinical data |
| ☒ | ☐ Dual use research of concern |
| ☒ | ☐ Plants |

## Methods

| n/a | Involved in the study |
|---|---|
| ☒ | ☐ ChIP-seq |
| ☒ | ☐ Flow cytometry |
| ☒ | ☐ MRI-based neuroimaging |

## Antibodies

| | |
|---|---|
| Antibodies used | Commerical antibodies used in the study: rabbit anti-RFP (Rockland immunochemicals, cat number 600-401-379), mouse anti-GFP (Sigma-Aldrich, cat number G1546), donkey anti-mouse IgG (H + L) IRDye® 800CW (LI-CORbioscience, cat number 926-32212) donkey anti-rabbit IgG (H + L) IRDye® 680CW (LI-CORbioscience, cat number 926-68072). Detailed information on commercial antibodies can also be found in the methods section |
| Validation | Commercial antibodies were validated in species mentioned above for immunofluorescence and western blots as noted on manufacturer's documentation and copied here.<br><br>Rabit anti-RFP validation according to manufactures website: This product was prepared from monospecific antiserum by immunoaffinity chromatography using Red Fluorescent Protein (Discosoma) coupled to agarose beads followed by solid phase adsorption(s) to remove any unwanted reactivities. Expect reactivity against RFP and its variants: mCherry, tdTomato, mBanana, mOrange, mPlum, mOrange and mStrawberry. Assay by immunoelectrophoresis resulted in a single precipitin arc against anti-Rabbit Serum and purified and partially purified Red Fluorescent Protein (Discosoma).  No reaction was observed against Human, Mouse or Rat serum proteins.<br><br>mouse anti-GFP validation: The isotype is determined using a double diffusion immunoassay using Mouse Monoclonal Antibody Isotyping Reagents,  Cat. Nos. G1546, ISO2.<br><br>secondaries validation: The antibody was isolated by affinity chromatography using antigens coupled to agarose beads. Based on ELISA, this antibody reacts with the heavy and light chains of mouse IgG, and with the light chains of mouse IgM and IgA. This antibody was tested by ELISA and/or solid-phase adsorbed to ensure minimal cross-reactivity with bovine, chicken, goat, guinea pig, horse, human, rabbit, and sheep serum proteins, but may cross-react with immunoglobulins from other species. The conjugate has been specifically tested and qualified for Western blot and In-Cell Western™ Assay applications. |

## Eukaryotic cell lines

Policy information about cell lines and Sex and Gender in Research

| | |
|---|---|
| Cell line source(s) | Human embryonic kidney 239T (HEK293T) cells (Cat#CRL-3216) were obtained from ATCC |
| Authentication | ATCC performs short-tandem repeat profiling for cell line authentication, and no additional cell line authentication |
| Mycoplasma contamination | The cell line was routinely checked for mycoplasma contamination using LT07-518 Mycoalert assay and has been verified as mycoplasma free. |
| Commonly misidentified lines (See ICLAC register) | The cell line used is not present in the list of commonly misidentified lines. |

## Plants

| | |
|---|---|
| Seed stocks | n/a |
| Novel plant genotypes | n/a |
| Authentication | n/a |

