## [Peer Review File · Nature Structural & Molecular Biology]

A network of interacting ciliary tip proteins with opposing activities imparts slow and processive microtubule growth

Corresponding Author: Professor Anna Akhmanova

Version 0:

Decision Letter:

11th Jun 2024

Dear Professor Akhmanova,

Thank you again for submitting your manuscript "A network of interacting ciliary tip proteins with opposing activities imparts slow and processive microtubule growth". We now have comments (below) from the 3 reviewers who evaluated your paper. In light of those reports, we remain interested in your study and would like to see your response to the comments of the referees, in the form of a revised manuscript.

You will see that, though the experts appreciate the potential novelty imparted by the study, they voice important concerns that need to be convincingly addressed in the form of a revised manuscript. More specifically, Reviewer #2 requests for the findings to be confirmed in a physiologically relevant setting, by investigating the effects of different stoichiometries and performing cellular experiments. We editorially agree that providing additional, convincing evidence about the robustness of the system and showing that it holds true in vivo, will be critical for the success of the story. Furthermore, Reviewer #2 questions the lack of localization specificity in vivo, raises questions about the cryo-ET data and statistical analyses, whereas Reviewer #1 asks for SEC to confirm the purity of the ciliary tip proteins and for a characterization of the CSPP1-ARMC9 interaction. Finally, all three reviewers ask for more analyses and some clarifications in the texts and figures.

Please be sure to address/respond to all concerns of the referees in full in a point-by-point response and highlight all changes in the revised manuscript text file.

We appreciate the requested revisions are extensive. We thus expect to see your revised manuscript within 6 months. If you cannot send it within this time, please let us know. We will be happy to consider your revision as long as nothing similar has been accepted for publication at NSMB or published elsewhere. Should your manuscript be substantially delayed without notifying us in advance and your article is eventually published, the received date would be that of the revised, not the original, version.

Reporting Summary:

When submitting the revised version of your manuscript, please pay close attention to our <https://www.nature.com/nature-portfolio/editorial-policies/image-integrity> Digital Image Integrity Guidelines and to the following points below:

- that unprocessed scans are clearly labelled and match the gels and western blots presented in figures.
- that control panels for gels and western blots are appropriately described as loading on sample processing controls.
- all images in the paper are checked for duplication of panels and for splicing of gel lanes.

Please ensure that you retain unprocessed data and metadata files after publication, ideally archiving data in perpetuity, as these may be requested during the peer review and production process or after publication if any issues arise.

Please note that all key data shown in the main figures as cropped gels or blots should be presented in uncropped form, with molecular weight markers. These data can be aggregated into a single supplementary figure. While these data can be displayed in a relatively informal style, they must refer back to the relevant figures. These data should be submitted with the last revision, prior to acceptance, but you may want to start putting it together at this point.

We urge authors to provide, in tabular form, the data underlying the graphical representations used in figures. This is to further increase transparency in data reporting, as detailed in this editorial (<http://www.nature.com/nsmb/journal/v22/n10/full/nsmb.3110.html>). Spreadsheets can be submitted in excel format. Only one (1) file per figure is permitted; thus, for multi-paneled figures, the source data for each panel should be clearly labeled in the Excel file; alternately the data can be provided as multiple, clearly labeled sheets in an Excel file. When submitting files, the title field should indicate which figure the source data pertains to. We encourage our authors to provide source data at the revision stage, so that they are part of the peer-review process.

We require deposition of coordinates (and, in the case of crystal structures, structure factors) into the Protein Data Bank with the designation of immediate release upon publication (HPUB). Electron microscopy-derived density maps and coordinate data must be deposited in EMDb and released upon publication. Deposition and immediate release of NMR chemical shift assignments are highly encouraged. Deposition of deep sequencing and microarray data is mandatory, and the datasets must be released prior to or upon publication. To avoid delays in publication, dataset accession numbers must be supplied with the final accepted manuscript and appropriate release dates must be indicated at the galley proof stage. Please find the complete NRG policies on data availability at <http://www.nature.com/authors/policies/availability.html>.

Please use the link below to submit your revised manuscript and related files:
Link Redacted

Sincerely,

Jean

Jean Nakhle, PhD
Associate Editor
Research Cross-Journal Editorial Team for Nature Structural & Molecular Biology
ORCID: 0000-0001-9385-6577

Reviewers' Comments:

Reviewer #1:

Remarks to the Author:

Saunders et al. reported biochemical and structural analyses of ciliary proteins, focusing on their effects on the MT dynamics. The authors discovered a slow yet steady-growing state when all five ciliary proteins were present, cryoET analyses showed a cork-like structure at the plus end that reduces protofilament flaring.

The discovery is novel and brings new insights into the long-standing questions about how ciliary MT grows. The TIRF MT assays and analyses are of high quality. The cryoET results are interesting but may need further clarification.

Point-by-point comments:

Introducing all these ciliary tip proteins presents a challenging task for the authors and the readers. As a reader, I am wondering how reliable are previous results of these ciliary tip interacting proteins. The evidence/understanding of CEP104, CSPP1, TOGARAM1, ARMC9, and CCDC66 may be at completely different levels and it's hard to gauge when the audience first reads the paper. Maybe a more thorough introduction of our previous understanding would help, but still challenging to write as the authors.

For protein purification, showing gels like Figure S1a is nice but not sufficient to show that these proteins are well-behaved. Something like SEC may exclude the possibility of aggregation, which can happen in the biochemistry of MT-binding proteins. Also, SEC would be helpful to tell if CEP104 is a dimer or not.

It was not clear what is the SKNN double mutation in "Indeed, CEP104 SKNN double mutation prevented EB3-dependent".

What is the interaction between CSPP1 and ARMC9? The authors did not mention if the effects shown in Fig. 5e and Fig. 5f depend on specific domains or residues (protein-protein interaction). The reviewer appreciated that in most cases, the authors did a good job including such experiments/results but it seems arbitrary to ignore them here.

It is quite striking to observe the slow growth of MT in the presence of CEP104 and TOGARAM1 (Fig. 6a). The reviewer recommends adding a few more kymographs showing that in the supplementary.

Similarly, when all five ciliary proteins were added, could the authors show more examples of the slow growth at the plus end?

It was unclear to the reviewer for the following two statements:

"It proved difficult to segregate all individual protofilaments from each other for further analysis due to the limited tilt range"

"We turned to manual tracing to analyze how the ciliary tip module may be altering MT tip architecture"

If the missing wedge is a problem, how could the authors manually trace the protofilaments?

It may be tough to track the protofilaments at specific orientations – how does that affect the quantification shown in Fig. 7k-o? How many MTs are analyzed? How many protofilaments are analyzed?

Reviewer #2:

Remarks to the Author:

In this manuscript, the authors used *in vitro* reconstitution of microtubule and a few known cilia tip proteins CEP104, CSPP1, TOGARAM1, ARMC9 and CCDC66 to get insight into the cilia assembly. With extensive tests of different constructs using truncations of different parts and pulldown assays, they confirmed and clarified the interactions between different tip proteins. Then, they used single-molecule microscopy to probe how each tip protein above influences microtubule dynamics. Furthermore, they used a combination of those proteins for single molecule microscopy and cryo-electron tomography (cryo-ET) to clarify the function and morphology of the cilia tip module proteins on microtubule growth. The microtubule dynamic assay shows that tip proteins stabilize the microtubule. CEP104 and CSPP1 act as pausers, and others act as microtubule elongation factors. The cryo-ET data showed that the tip module proteins form cork-like structures at the plus-end of the microtubule and affect the flare of the microtubule plus-end. They concluded that the cilia tip module proteins stabilize the microtubule and allow slow progressive microtubule elongation, which resemble the axonemal assembly in cells.

While I appreciate the extensive *in vitro* work of the work with many constructs of tip proteins expressed, purified and tested and the thorough presentation and sharing of the data, I am not totally convinced about the significance of the manuscript with the interpretation of *in vitro* reconstitution to what happens in the cells.

Here are my concerns:

Throughout the manuscript, some of the tip proteins did not just localize to the plus-end but some islands/spots in the microtubules or along the length of the microtubule (with/without EB3). For example, CCDC66 binds along the newly growing microtubule, CEP104 does not always bind to the tip (Fig. 1L, Fig. 3F, G), CSPP1 binds to a lot of places along the length of dynamic microtubules, TOGARAM1 binds stronger in the microtubule seed (Fig. 1o, p). *In vivo* data shows that these proteins are more enriched at the tip (or some also at the proximal region of the cilia).

Even in the case of the full ciliary tip module (Fig. 6), the signals from CEP104 did not localize to the tip specifically (with or without CSPP1). The localization without TOGARAM1 ARMC9 or CEP104 or CCDC66 is reasonably different, which makes the interpretation not easy to explain.

When reconstituting the tip module, the authors also tried to do TIRF by omitting a component, one by one and compared to the full module. That allows them to talk about the role of individual proteins. However, I believe the more relevant experiment is to change the stoichiometry of each component to see whether it can affect the dynamic state significantly because omitting one component leads to different localization of the tip proteins as shown in Fig. 6i to n. There is no

justification for the stoichiometry used relative to what is found in vivo. Therefore, linking this specific stoichiometry and omission of components might not provide a good link to what happens in vivo.

I also have concerns regarding the cryo-ET data of the full ciliary tip module. I looked through the uploaded tomograms, and unfortunately, there were no labels on which one is with/without the module. Therefore, I have to guess based on the appearance of the modules and assume that it is the tomogram with the modules.

- There are no statistics on how many reconstituted microtubules in the presence of the modules have the cork and without the cork. In those tomograms showing microtubule and the cork densities, I did notice some without the cork. For example, in tomo22, there was one MT that was very similar to dynamic MT at the bottom and one with a flare tip on top and no obvious density inside. In tomo38, tomo52 and 64, there is one MT without density, with a flared end. It is unclear to me whether it is determined to be the minus end. These statistics can be done by 2D imaging (no tilt series) due to the obvious appearance of the champagne cork density. With 2D imaging, many microtubules can be imaged and categorized.

- Perhaps, the minus-end can also be quantified for dynamic MT and MT + ciliary tip module in the supplementary to clarify the statistic as well.

- In addition, the microtubules with the tip module look very clean, while TIRF imaging clearly shows that many tip proteins bind outside the tip region (Fig. 6). Should there be density outside microtubules explaining the island of protein outside the tip observed in Fig. 6 to be included?

- Also, the appearance of the cork densities do not support the biochemical data fully. Out of all the 5 tip proteins, CSPP1 is a luminal protein. CEP104, TOGARAM1, and CCDC66 are outside proteins and ARMC9 is suggested not to bind MT in their data while it contains a TOG and LisH domain, which normally binds MT. However, the majority of the density is toward the inside. There are not many densities binding outside to explain the MTBD of CEP104, TOGARAM1 and CCDC66. The molar concentrations of CSPP1:CEP104:TOGARAM1:CCDC66:ARMC9 in the experiment are 20:20:20:20:200 nM. The authors stated that for several reasons, we do not think the five ciliary tip proteins form a stoichiometric complex but rather a flexible interaction network, the function of which is likely to enhance the accumulation of these proteins at the distal ends of cilia. However, it is hard to imagine most of the proteins are luminal.

Reviewer #3:

Remarks to the Author:

In this manuscript, Saunders and colleagues describe the opposing activities of ciliary tip member complex in vitro that regulate axonemal microtubule growth in vivo. The authors focus on the complex members CEP104, CSP1, TOGARAM1, ARMC9 and CCDC66, whose mutations have been associated to Joubert syndrome, and analyze their effect individually or in combination on microtubule dynamics. The overarching goal of this work is to uncover the biochemical mechanisms underlying the function of these proteins and their combined regulation of ciliary MTs since this process is currently poorly understood. Indeed, axonemal MTs do not have a stable GTP cap but instead have specific ciliary module associated with their plus end which regulate their growth. How is this regulation achieved is the focus of this work. To tackle this question, the authors use a combination of biochemistry methods, TIRF microscopy and cryo-ET.

Importantly, the authors could disentangle the action of each component individually as well as in combination in regulating MT growth (as a proxy for axonemal growth, which is microtubule doublets based). They notably describe the action of CEP104 as inhibiting MT plus end elongation and shortening, an activity that is potentiated by EB3, CSPP1 and CCDC66. In contrast, the other TOG-containing protein TOGARAM1 converts this inhibition into slow growth, while ARMC9 and CCDC66 act as scaffold proteins. Finally, they found that the ciliary module forms a cork like structure at MT plus end and diminishes protofilament flaring. Altogether, the authors show elegantly that this module, through a combination of opposing activities, stabilize MT plus ends and drive slow elongation.

To my opinion, this is an excellent manuscript, well written, whose conclusions are based on a thorough study of the underlying mechanisms of the members of the ciliary tip complex on MT dynamics. The quality of the work is very high, and the results are very important for our understanding of this very specific slow MT growth observed not only in cilia but also on their associated structure, the centriole.

Overall, this manuscript provides a very nice dissection of the tip module's mechanism of action, with very well-described interactions. The work is clean, with all controls present, beautiful and very well done with appropriate statistical analyses and will be useful to a wide target audience such as the one of NSMB, especially for those working on MTs, cilia and centrioles.

I only have a small number of minor comments that need to be addressed before publication.

1-Page 5: the authors mention that TOGARAM1 decreases MT growth rate and that this is unexpected but do not comment more on this? Could they try to expand a bit on this?

2- the data is very convincing. For the graphs Figure 1d, e, f, the labels are only in f (CCDC66, etc) which makes it a bit difficult to read. I know that the space is limiting but may be boxing the lines of the column bars with a dedicated color for the

protein may help (might also be too many colors to be tested, may be use patterns instead of colors for the pause to shrinkage, etc). could the authors try to make the reading on the graph easier if possible?

3-FRAP of CEP104 shows no recovery (Fig 2c)- CEP104 does not exchange at blocked MTs plus end. What about CEP104 FRAP experiment in the context of the full module (the five proteins together as in Fig 6/7)? Would you see some recovery since the authors propose that the tip complex form a flexible interaction network? Or do you think that CEP104 will still not exchange under these conditions and stays at the MT tip?

4-This is more a discussion point: The authors convincingly show how the tip complex can modulate slow axonemal MT growth. They mention that some of these proteins are at centrioles. Do they think such a complex (like the CP110/CPAP one which also display a "plug" configuration) would be at act also during centriole assembly ensuring a very slow growth of the procentriole? Which members of the complex do not localize at centrioles? Can the authors extend the discussion of this point further to what is already written. Do they think that CEP104 will also be in the CP110 complex at centriole since the two proteins can interact? Or do they think that specific complexes exist to perform similar function at centrioles and at the plus end of axoneme?

Version 1:

Decision Letter:

Our ref: NSMB-A49153A

15th Nov 2024

Dear Dr. Akhmanova,

Thank you for submitting your revised manuscript "A network of interacting ciliary tip proteins with opposing activities imparts slow and processive microtubule growth" (NSMB-A49153A). It has now been seen by the original referees and their comments are below. The reviewers find that the paper has improved in revision, and therefore we'll be happy in principle to publish it in Nature Structural & Molecular Biology, pending minor revisions to satisfy the referees' final requests and to comply with our editorial and formatting guidelines.

We are now performing detailed checks on your paper and will send you a checklist detailing our editorial and formatting requirements in about 2-3 weeks. Please do not upload the final materials and make any revisions until you receive this additional information from us.

To facilitate our work at this stage, it is important that we have a copy of the main text as a word file. If you could please send along a word version of this file as soon as possible, we would greatly appreciate it; please make sure to copy the NSMB account (cc'ed above).

Sincerely,

Katarzyna Ciazynska, PhD
(she/her)
Senior Editor
Nature Structural & Molecular Biology
<https://orcid.org/0000-0002-9899-2428>

Reviewer #2 (Remarks to the Author):

The authors addressed all of my concerns. It's good to put down the statistics of cryoET in so the audience know what were analyzed.

This in vitro work is the first step towards understanding the functions of so many ciliary MT-binding proteins. While I agree with the other reviewer that in vitro may not recapitulate everything that happens in cells, I still think this in vitro work represents the state-of-the-art understanding and provides new hypotheses to be tested. It also fills a long-lasting gap in the field of ciliary biochemistry.

Reviewer #3 (Remarks to the Author):

In this revision, the authors addressed the concerns and suggestions of reviewers (me and two other reviewers) with more experiments and analysis, such as quantifying the cork density in the cryo-ET and stoichiometry variation in the tip proteins for the microtubule dynamic assays. They also added texts in the results and discussions to clarify different points.

While there are still things unclear about the cryo-ET data, such as the microtubule service is very clean without many densities binding outside, the cryo-ET is only a part of the manuscript. Therefore, I believe the author addressed the concern appropriately in this revision. The paper is good for publication for their findings.

A small suggestion.

In methods, the authors need to show how many tomograms were collected and if different datasets are collected (biological replicate or technical replicate). This will support the reproducibility as well.

Furthermore, if possible, the cryo-ET data or at least representative data should be deposited or shared on EMPIAR or EMDB.

Reviewer #4 (Remarks to the Author):

The authors responded satisfactorily to the points raised during the review process. The new provided results are convincing. I am convinced by the revised version of the manuscript and recommend it for publication in NSMB.

Version 2:

Decision Letter:

3rd Jan 2025

Dear Dr. Akhmanova,

We are now happy to accept your revised paper "A network of interacting ciliary tip proteins with opposing activities imparts slow and processive microtubule growth" for publication as a Article in Nature Structural & Molecular Biology.

Your paper will be published online soon after we receive proof corrections and will appear in print in the next available issue. You can find out your date of online publication by contacting the production team shortly after sending your proof corrections.

You may wish to make your media relations office aware of your accepted publication, in case they consider it appropriate to organize some internal or external publicity. Once your paper has been scheduled you will receive an email confirming the publication details. This is normally 3-4 working days in advance of publication. If you need additional notice of the date and time of publication, please let the production team know when you receive the proof of your article to ensure there is sufficient time to coordinate. Further information on our embargo policies can be found here:

<https://www.nature.com/authors/policies/embargo.html>

Kind regards,
Florian

Dr Florian Ullrich
Senior Editor, Nature
Consulting Editor, Nature Structural & Molecular Biology
ORCID 0000-0002-1153-2040

Point-by-point answers to the Reviewers' Comments:

Reviewer #1:

Remarks to the Author:

Saunders et al. reported biochemical and structural analyses of ciliary proteins, focusing on their effects on the MT dynamics. The authors discovered a slow yet steady-growing state when all five ciliary proteins were present, cryoET analyses showed a cork-like structure at the plus end that reduces protofilament flaring.

The discovery is novel and brings new insights into the long-standing questions about how ciliary MT grows. The TIRF MT assays and analyses are of high quality. The cryoET results are interesting but may need further clarification.

We thank the reviewer for their supportive and insightful comments. We have addressed their concerns clarifying cryo-ET results and other data, as described in detail below.

Point-by-point comments:

Introducing all these ciliary tip proteins presents a challenging task for the authors and the readers. As a reader, I am wondering how reliable are previous results of these ciliary tip interacting proteins. The evidence/understanding of CEP104, CSPP1, TOGARAM1, ARMC9, and CCDC66 may be at completely different levels and it's hard to gauge when the audience first reads the paper. Maybe a more thorough introduction of our previous understanding would help, but still challenging to write as the authors.

We have extended and modified the Introduction to clarify how the data were obtained, and where relevant, also indicated the species.

For protein purification, showing gels like Figure S1a is nice but not sufficient to show that these proteins are well-behaved. Something like SEC may exclude the possibility of aggregation, which can happen in the biochemistry of MT-binding proteins. Also, SEC would be helpful to tell if CEP104 is a dimer or not.

SEC-MALS analysis of purified proteins has now been included. We see that none of the ciliary tip module proteins aggregate. The following figure (new Figure S1b) and the accompanying text (pages 5 and 8 of the revised manuscript) have been added to the manuscript.

New Figure S1b. SEC-MALS analysis of purified GFP-tagged ciliary tip module proteins. The human sequence was used for all module members, except for TOGARAM1. Due to technical difficulties, the mouse sequence of TOGARAM1 was used, which is 84% identical to human TOGARAM1.

Results section, page 5:

“Size exclusion chromatography detected no protein aggregation, and mass spectrometry-based analysis demonstrated that the protein preparations contained only very minor contaminations with other known regulators of MT dynamics, such as CSPP1 (Fig. S1b, c).”

Page 8:

“Size exclusion chromatography revealed that ARMC9 forms multimers (Fig. S1b)”

Our priority with size exclusion chromatography experiments was to establish if any of the ciliary tip proteins formed aggregates, and for this reason we chose the resin best suited to resolve these differences. It was therefore not possible to distinguish if CEP104 is a dimer from these data; furthermore, we think that the GFP counting data reported in Figure 2a,b, are more reliable, because they are obtained in the same conditions as our in vitro reconstitution assays.

It was not clear what is the SKNN double mutation in “Indeed, CEP104 SKNN double mutation prevented EB3-dependent”.

The text has been changed to address this comment as follows (results section, page 6):

“This could be explained by the direct interaction of EB3 and CEP104 on the MT through the SxIP motif of CEP104, SKIP (Fig. 3a) ¹⁶. Mutation of this motif has previously been shown to abolish the interactions between the C-terminus of EB and its partners ^{31, 46}. Indeed, double mutation of CEP104’s SxIP motif, SKIP to SKNN, prevented EB3-dependent recruitment of CEP104 to MTs; therefore, even in the presence of EB3, 100 nM CEP104 SKNN was needed to block plus-end growth (Fig. 3a-d, and S2a).”

What is the interaction between CSPP1 and ARMC9? The authors did not mention if the effects shown in Fig. 5e and Fig. 5f depend on specific domains or residues (protein-protein

interaction). The reviewer appreciated that in most cases, the authors did a good job including such experiments/results but it seems arbitrary to ignore them here.

We thank the reviewer for pointing out this omission. Additional in vitro reconstitution experiments have been carried out with ARMC9 and CSPP1 constructs (New Fig. 5g, h and S5c). These experiments confirmed that the C-terminal portion of CSPP1 was required to recruit ARMC9 to microtubules and showed that ARMC9 does not alter the impact of CSPP1 deletion mutants on microtubule dynamics. These additional data have been included below and in the text on pages 9-10.

New Fig 5 g, h. Fields of view (left, scale bar 2 μ m) and kymographs (right, scale bars 2 μ m and 60 s) illustrating MT dynamics from GMPCPP-stabilized seeds with 20 nM GFP-EB3 and indicated concentrations and colors of CSPP1 constructs and ARMC9. Yellow arrowheads, colocalization.

New Fig S5c. Fields of view (left, scale bar 2 μm) and kymograph (right, scale bars 2 μm and 60 s) illustrating MT dynamics from GMPCPP-stabilized seed with GFP-EB3 and indicated concentrations and colors of CSPP1 constructs.

Accompanying text in results section, pages 9-10:

“Through co-immunoprecipitation experiments using CSPP1 and ARMC9 truncations, we mapped this interaction to the C-terminal part of CSPP1 and the linker regions surrounding the central helical domain of ARMC9 (Fig. S4a, S5a and b). When tested in in vitro reconstitution experiments, we observed that, as previously reported, the shorter CSPP-S construct was less potent at pausing MTs, whereas CSPP-MTOR (MT-organizing region) induced relatively few pauses but was promote rescue events (Fig. S5c) 15, 17, 34, and these effects were not changed by the addition of ARMC9 (Fig. 5g, h). The C-terminal portion of CSPP1 was required to recruit ARMC9 to MTs, confirming our co-immunoprecipitation experiments (Fig. 5g, h and S5c).”

It is quite striking to observe the slow growth of MT in the presence of CEP104 and TOGARAM1 (Fig. 6a). The reviewer recommends adding a few more kymographs showing that in the supplementary.

Additional data have been included in the new Figure S6c.

New figure S6c. Additional kymographs (scale bars 2 μm and 60 s) illustrating MT dynamics from GMPCPP-stabilized seed with GFP-EB3, mCherry-TOGARAM1, and GFP-CEP104. Purple arrows, blocked plus end; green arrows, slow MT growth.

Similarly, when all five ciliary proteins were added, could the authors show more examples of the slow growth at the plus end?

Additional examples were included in the new Figure S6e.

New figure S6e. Additional kymographs (scale bars 2 μm and 60 s) illustrating MT dynamics from GMPCPP-stabilized seeds of the entire ciliary tip module with 20 nM GFP-EB3. Green arrows, slow MT growth.

It was unclear to the reviewer for the following two statements:

“It proved difficult to segregate all individual protofilaments from each other for further analysis due to the limited tilt range”. “We turned to manual tracing to analyze how the ciliary tip module may be altering MT tip architecture” If the missing wedge is a problem, how could the authors manually trace the protofilaments?

It may be tough to track the protofilaments at specific orientations – how does that affect the quantification shown in Fig. 7k-o? How many MTs are analyzed? How many protofilaments are analyzed?

We thank the reviewer for highlighting potential points of confusion. We have modified the text to clarify that we did manual protofilament tracing as a better alternative for determining single protofilament parameters (Results section, page 12):

“To visualize MT protofilaments and ciliary tip module, we initially performed semi-automated segmentation of the tomographic volumes (Fig. 7d and e). However, using this approach, it was difficult to distinguish individual protofilaments for further analysis. We therefore turned to manual protofilament tracing (see Materials and Methods) and were able to trace the majority (~13) protofilaments of each MT end (Fig. 7f and g, S7a and b).”

In addition, we added the following text to the Materials & Methods section (page 24) for additional clarification in the missing wedge:

“Although the missing wedge stretching complicated resolving neighboring protofilaments, most protofilaments could be distinguished with manual tracing due to differences in length or their deviation from the plane parallel to the imaging direction (because of flaring). Additionally, the total protofilament number of MT tips was determined prior to tracing, using 2D rotational symmetry averaging of the MT cross-section. We looked for neighboring protofilaments at angular distances corresponding to the determined protofilament number ($360^\circ/\text{PF number}$) with an error margin of $\pm 4^\circ$ to account for potentially imperfect tilt alignments of tomograms. Furthermore, to faithfully trace protofilaments at specific orientations,

the tracing procedure was monitored from orthogonal views using 3dmod zap and slicer windows, as well as the volume viewer to show the electron density signal in 3D. Nonetheless, due to the signal-to-noise ratio in the tomograms, we cannot exclude the possibility of slight inaccuracies in the manual tracing, especially with extra densities present from the ciliary tip module. Protofilaments that could not be confidently traced were excluded from the analysis. The number of analyzed MTs and protofilaments can be found in the figure legend of Fig. 7.”

A total of 329 protofilaments from 26 microtubules were analyzed for controls, and 441 protofilaments from 33 microtubules for ciliary tip module samples, averaging approximately 13 protofilaments from each microtubule.

Reviewer #2:

Remarks to the Author:

In this manuscript, the authors used in vitro reconstitution of microtubule and a few known cilia tip proteins CEP104, CSPP1, TOGARAM1, ARMC9 and CCDC66 to get insight into the cilia assembly. With extensive tests of different constructs using truncations of different parts and pulldown assays, they confirmed and clarified the interactions between different tip proteins. Then, they used single-molecule microscopy to probe how each tip protein above influences microtubule dynamics. Furthermore, they used a combination of those proteins for single molecule microscopy and cryo-electron tomography (cryo-ET) to clarify the function and morphology of the cilia tip module proteins on microtubule growth. The microtubule dynamic assay shows that tip proteins stabilize the microtubule. CEP104 and CSPP1 act as pausers, and others act as microtubule elongation factors. The cryo-ET data showed that the tip module proteins form cork-like structures at the plus-end of the microtubule and affect the flareness of the microtubule plus-end. They concluded that the cilia tip module proteins stabilize the microtubule and allow slow progressive microtubule elongation, which resemble the axonemal assembly in cells.

While I appreciate the extensive in vitro work of the work with many constructs of tip proteins expressed, purified and tested and the thorough presentation and sharing of the data, I am not totally convinced about the significance of the manuscript with the interpretation of in vitro reconstitution to what happens in the cells.

The key point of our paper is that combined, ciliary tip proteins impart slow processive tubulin addition to microtubule plus ends, mimicking slow microtubule growth occurring at the tips of cilia. Obviously, our reconstitutions will not recapitulate all aspects of the localization and function of the studied proteins, but they do provide insight into the individual and collective biochemical activities of these proteins on dynamic microtubules, which have until now have not been explored.

Here are my concerns:

Throughout the manuscript, some of the tip proteins did not just localize to the plus-end but some islands/spots in the microtubules or along the length of the microtubule (with/without EB3). For example, CCDC66 binds along the newly growing microtubule, CEP104 does not always bind to the tip (Fig. 1L, Fig. 3F, G), CSPP1 binds to a lot of places along the length of dynamic microtubules, TOGARAM1 binds stronger in the microtubule seed (Fig. 1o, p). In vivo data shows that these proteins are more enriched at the tip (or some also at the proximal region of the cilia).

Most of the studied proteins are indeed not microtubule tip-specific in vitro, but, as outlined below, this is also often true for these proteins in cells.

1. CCDC66: a very recent paper (published after our initial submission) showed that endogenous CCDC66 localizes along the axoneme of primary cilia in IMCD3 cells, and in fact, the authors of this study do not see any enrichment at the ciliary tip (Deretic *et al.*, 2024. doi:10.1101/2024.06.16.599243).
2. CSPP1: this protein is known to be a microtubule luminal protein, it is therefore possible that antibody stainings would primarily see localization only at ciliary tips where CSPP1 is likely most accessible. Notably, in transfection experiments in RPE1 cells mNG-CSPP1 was observed to be enriched at both ciliary tips and along the transition zone, and additionally it was also seen to weakly decorate the length of cilia (Frikstad *et al.*, 2019, doi: 10.1016/j.celrep.2019.07.025).
3. CEP104: There is currently no commercial antibody available against CEP104 that successfully works in cells. Whilst expression of tagged CEP104 has shown it to primarily localize at ciliary tips, when co-transfected with CSPP1, CEP104 localizes to spots along the axoneme as well as at ciliary tip (Frikstad *et al.*, 2019, doi: 10.1016/j.celrep.2019.07.025). Furthermore, it is known that CEP104 can bind to EB1 and track growing microtubule ends in cells in EB-dependent manner (Jiang *et al.*, 2012, doi: 10.1016/j.cub.2012.07.047). CEP104-EB interaction was recapitulated in our experiments: EB3 enhanced plus end blocking by CEP104, and the two proteins could track together growing microtubule minus ends (see, for example, Figure 1l).
4. TOGARAM1: Endogenous staining of mouse TOGARAM1 in IMCD3 cells as well as transient transfection in HEK-293 cells shows localization along cilium and at basal bodies (Das *et al.*, 2015, doi: 10.1091/mbc.e15-08-0603).

It is also important to mention that in vitro assays are the only way to find out whether a particular protein autonomously recognizes microtubule ends, either dynamic or stable ones. As to the seed localization (e.g. observed with TOGARAM1), since seeds are stabilized by GMPCPP, they mimic some aspects of microtubule tip structure, such as expanded lattice conformation, and it is not unusual that some proteins with a preference for microtubule tips also bind to the seeds. In fact, such information can be helpful for teasing out the preferred binding sites of different proteins on the microtubule lattice.

Even in the case of the full ciliary tip module (Fig. 6), the signals from CEP104 did not localize to the tip specifically (with or without CSPP1). The localization without TOGARAM1, ARMC9 or CEP104 or CCDC66 is reasonably different, which makes the interpretation not easy to explain.

When reconstituted with the entire module, we always see CEP104 at microtubule tips, although it often also localizes to the microtubule seed in these conditions (Fig. 6j, m-o). We believe this is the effect of binding to TOGARAM1. As the reviewer previously mentioned, TOGARAM1 is enriched on microtubule seeds, and as we showed that CEP104 and TOGARAM1 interact, we believe it is this interaction that is

also recruiting CEP104 to the seed. This can be most clearly seen in assays with just TOGARAM1 and CEP104 where CEP104 is also present on the seed (Fig 6a and S6c). Furthermore, in Figure 6k when TOGARAM1 was excluded from the ciliary tip module reconstitution, CEP104 was no longer present at the seed vs Figure 6j of the entire tip module or Figure 6m-o of the 4 member tip module reconstitutions with TOGARAM1 where CEP104 is present at the seeds.

Accompanying results text, page 10:

“When CEP104 and TOGARAM1 were combined on dynamic MTs in the presence of EB3, CEP104 was located not only at the plus ends, but also at MT seeds, likely due to the binding of CEP104 to TOGARAM1, which is enriched at MT seeds (Fig. 1l, p, 6a and S6c).”

We have added the CEP104 channel to Fig 6j-o:

Figure 6i-o. Reconstitutions of indicated concentrations and colors of ciliary tip module proteins (i). Fields of view (left, scale bar 2 μm) and kymographs (j-o) (middle,

combined colors; right, CEP104 only; scale bars 2 μm and 60 s) illustrating MT dynamics from GMPCPP-stabilized seeds with 20 nM GFP-EB3 (in magenta) and ciliary tip module components indicated in (i).

When reconstituting the tip module, the authors also tried to do TIRF by omitting a component, one by one and compared to the full module. That allows them to talk about the role of individual proteins. However, I believe the more relevant experiment is to change the stoichiometry of each component to see whether it can affect the dynamic state significantly because omitting one component leads to different localization of the tip proteins as shown in Fig. 6i to n. There is no justification for the stoichiometry used relative to what is found in vivo. Therefore, linking this specific stoichiometry and omission of components might not provide a good link to what happens in vivo.

Many Joubert syndrome mutations result in a complete loss of protein, and therefore, leaving out one protein at a time is relevant, because it is a biochemical mimic of a genetic deficiency. However, titrating the levels of each protein within the complex is also an excellent suggestion, especially as protein stoichiometry in vivo is unknown, and this also helps to shed light on the robustness of the in vitro reconstitution results. For the titration experiments, we focused on the three proteins that directly modulate microtubule dynamics, TOGARAM1, CEP104, and CSPP1, because when we removed either ARMC9 or CCDC66 individually, we saw very little effect on the percentage of time microtubules spent growing slowly (new Fig. 6h) and no change in CEP104 localization (Fig. 6n and o). Interestingly, when we left out both ARMC9 and CCDC66, pausing factors became more dominant: at 10 nM TOGARAM1, CEP104 and CSPP1, all microtubule plus ends were blocked. Slow growth was regained in the absence of both ARMC9 and CCDC66 when the concentrations of CEP104 and CSPP1 was reduced, while a further relative increase in TOGARAM1 concentration led to the appearance of episodes of fast growth. Interestingly, although modulating the concentrations of TOGARAM1, CEP104, and CSPP1 affected the amount of time microtubules spent paused or growing (either rapidly or slowly), we were surprised to see that we could not modulate the rate of slow growth at any of the ratios we tested. We conclude that slow growth is quite a robust state, which is not particularly sensitive to the changes in protein stoichiometry, which, however, affects the frequency of slow growth episodes. The quantification of this new experiment has been added below and to Figure 6h and p with kymograph examples included in Figure S6g-k and accompanying text on page 11.

Figure 6h, Dynamic states for combinations of ciliary tip module proteins indicated (from kymographs j-o, 1c, and S6g-k). Bars represent or averaged means from two independent experiments, error bars represent s.e.m..

New figure 6p, weighted growth rates for combinations of indicated ciliary tip module proteins (from kymographs S6g-k). Data was pooled from three independent experiments, total number of growth events: 10 nM TOGARAM1 with 10 nM CEP104 and 10 nM CSPP1, n=26; 10 nM TOGARAM1 with 2 nM CEP104 and 2 nM CSPP1, n=58; 20 nM TOGARAM1 with 2 nM CEP104 and 2 nM CSPP1, n=88; 20 nM TOGARAM1 with 2 nM CEP104 and 0.5 nM CSPP1, n=77; 20 nM TOGARAM1 with 0.5 nM CEP104 and 2 nM CSPP1, n=228.

New Figure S6g-k, Fields of view (left, scale bar 2 μm) and kymographs (right, scale bars 2 μm and 60 s) illustrating MT dynamics from GMPCPP-stabilized seeds with 20 nM GFP-EB3 and indicated concentrations and colors of ciliary tip module proteins.

Accompanying text, Results page 11:

“Next, we removed ARMC9 and CCDC66 simultaneously, and were surprised to see that in these conditions, with TOGARAM1, CEP104 and CSPP1 at 10 nM each, almost all MT plus ends were paused (Fig. 6h and S6g). However, when we reduced the concentration of CEP104 and CSPP1 to 2 nM in the presence of 10 nM TOGARAM1, episodes of slow growth with an average rate of $0.022 \pm 0.01 \mu\text{m}/\text{min}$ were again observed (Fig. 6h, p and S6h). Whilst raising TOGARAM1 concentration to 20 nM further increased the percentage of the time MTs spent growing slowly, the rate of these slow growth episodes remained essentially the same (Fig. 6h, p and S6i). Slow growth rate remained relatively constant also when either CEP104 or CSPP1 concentration was decreased to 0.5 nM, but periods of fast growth were again observed in these conditions (Fig. 6h, p and S6j, k). Together, these results demonstrate that slow growth is a relatively robust state that is triggered at a certain ratio of TOGARAM1 and the two pausing factors. When the latter are present in excess, the scaffold proteins ARMC9 and CCDC66 can stimulate TOGARAM1-dependent slow growth. Altogether, ciliary tip module components stabilize MTs by preventing their depolymerization and promoting robust but very slow tubulin addition at a rather constant rate.”

Accompanying text, Discussion page 13:

“The slow growth state of MT plus ends was robust as, interestingly, we could not modulate its rate by titrating protein concentrations, suggesting that it is not overly sensitive to protein stoichiometry.”

Discussion page 15:

“Furthermore, our observation that both ARMC9 and CCDC66 can potentiate slow MT growth when the pausing factors CEP104 and CSPP1 predominate over TOGARAM1 helps to explain how scaffolding proteins can promote cilia elongation.”

I also have concerns regarding the cryo-ET data of the full ciliary tip module. I looked through the uploaded tomograms, and unfortunately, there were no labels on which one is with/without the module. Therefore, I have to guess based on the appearance of the modules and assume that it is the tomogram with the modules.

We are sorry that the reviewer could not find any labels on the cryo-ET data; it seems like our folder structure unfortunately got lost when the journal shared the data with the reviewer. Therefore, we now label all the files individually.

- *There are no statistics on how many reconstituted microtubules in the presence of the modules have the cork and without the cork. In those tomograms showing microtubule and the cork densities, I did notice some without the cork. For example, in tomo22, there was one MT that was very similar to dynamic MT at the bottom and one with a flare tip on top and no obvious density inside. In tomo38, tomo52 and 64, there is one MT without density, with a flared end. It is unclear to me whether it is determined to be the minus end. These statistics can be done by 2D imaging (no tilt series) due to the obvious appearance of the champagne cork density. With 2D imaging, many microtubules can be imaged and categorized.*
- *Perhaps, the minus-end can also be quantified for dynamic MT and MT + ciliary tip module in the supplementary to clarify the statistic as well.*

We thank the reviewer for pointing out the missing statistics on the appearance of corks for both plus- and minus ends, therefore we have added a panel to Fig. 7 where we show this quantification based on our cryo-ET data:

New Figure 7c, Cork distribution in the ciliary tip module samples, based on visual inspection for the presence of cork densities and based on chirality of the rotationally

averaged cross-section for MT polarity. Total number of MT ends analyzed; plus ends, n = 60; minus ends, n = 39.

Modified text, Results page 12:

“85% of plus ends found in the ciliary tip module dataset, where the chirality of the cross-section could be resolved, were corked. Furthermore, all minus ends were free (Fig. 7c).”

As we had a large dataset already and wanted to use it to its full potential, this quantification is based on our existing tomography data as well as additional samples imaged with cryo-ET instead of the suggested 2D imaging. Furthermore, from the quantification based on this dataset, it is clear the in vitro reconstitution is robust and, in our opinion, does not require any further statistics that could be attained through 2D imaging.

- In addition, the microtubules with the tip module look very clean, while TIRF imaging clearly shows that many tip proteins bind outside the tip region (Fig. 6). Should there be density outside microtubules explaining the island of protein outside the tip observed in Fig. 6 to be included?

Indeed, this is an interesting observation that we did not fully explore previously. We have now imaged additional CTM samples with cryo-ET and found more microtubules with luminal densities located away from the tip, which reflects our TIRF observations. There seems to be some variability between samples with respect to the frequency of luminal densities, which may explain why we missed this previously. However, we were not able to find any clear densities at the outside of the MT. Exemplary images of the new data are shown in Fig S7 c and d:

New Figure S7c, d, Slices (4.3 nm thick) of denoised tomograms showing MTs grown in presence of EB3 alone (c) or the ciliary tip module and EB3 (d). Insets show a zoom of the indicated region. Scale bars 100 nm.

Modified text, Results page 12:

“At corked MT tips, extensive contacts on the luminal surface of tubulin and the exposed protofilament ends were observed. Luminal densities were also present at regions located away from the cork, however, we could not detect any visible electron densities at the outer MT surface (Fig. S7c and d).”

Data of the new ciliary tip module samples is also included in all our analyses. Therefore, old graphs in Fig 7 were adjusted, as additional tracing and related analyses were performed on 7 more corked microtubules.

- Also, the appearance of the cork densities do not support the biochemical data fully. Out of all the 5 tip proteins, CSPP1 is a luminal protein. CEP104, TOGARAM1, and CCDC66 are outside proteins and ARMC9 is suggested not to bind MT in their data while it contains a TOG and LisH domain, which normally binds MT. However, the majority of the density is toward the inside. There are not many densities binding outside to explain the MTBD of CEP104, TOGARAM1 and CCDC66. The molar concentrations of CSPP1:CEP104:TOGARAM1:CCDC66:ARMC9 in the experiment are 20:20:20:20:200 nM. The authors stated that for several reasons, we do not think the five ciliary tip proteins form a stoichiometric complex but rather a flexible interaction network, the function of which is

likely to enhance the accumulation of these proteins at the distal ends of cilia. However, it is hard to imagine most of the proteins are luminal.

We agree that finding predominantly luminal densities was somewhat unexpected. One reason could be that proteins bound to the outer microtubule surface are more difficult to see by cryo-ET, especially if they are partially unstructured or extend some distance away from the microtubule surface. For example, it is well-established that in cells, growing microtubule plus ends are decorated by +TIP comets which can be easily detected by immunofluorescence-based staining or live imaging, yet these comets are not obvious in published EM images of cells. We thus think it likely that the luminal part of the complex is more compact and therefore better visible in the conditions we used for cryo-ET. As a side note, ARMC9 contains a LisH and an Armadillo repeat domain (and not a TOG domain, see Figure 1a), neither of which by themselves are known to bind to microtubules, so it is not surprising that it does not bind to microtubules but rather acts as a scaffold, and since it binds to CSPP1, it likely adds to the luminal density observed by cryo-ET.

Reviewer #3:

Remarks to the Author:

In this manuscript, Saunders and colleagues describe the opposing activities of ciliary tip member complex in vitro that regulate axonemal microtubule growth in vivo. The authors focus on the complex members CEP104, CSP1, TOGARAM1, ARMC9 and CCDC66, whose mutations have been associated to Joubert syndrome, and analyze their effect individually or in combination on microtubule dynamics. The overarching goal of this work is to uncover the biochemical mechanisms underlying the function of these proteins and their combined regulation of ciliary MTs since this process is currently poorly understood. Indeed, axonemal MTs do not have a stable GTP cap but instead have specific ciliary module associated with their plus end which regulate their growth. How is this regulation achieved is the focus of this work. To tackle this question, the authors use a combination of biochemistry methods, TIRF microscopy and cryo-ET.

Importantly, the authors could disentangle the action of each component individually as well as in combination in regulating MT growth (as a proxy for axonemal growth, which is microtubule doublets based). They notably describe the action of CEP104 as inhibiting MT plus end elongation and shortening, an activity that is potentiated by EB3, CSPP1 and CCDC66. In contrast, the other TOG-containing protein TOGARAM1 converts this inhibition into slow growth, while ARMC9 and CCDC66 act as scaffold proteins. Finally, they found that the ciliary module forms a cork like structure at MT plus end and diminishes protofilament flaring. Altogether, the authors show elegantly that this module, through a combination of opposing activities, stabilize MT plus ends and drive slow elongation.

To my opinion, this is an excellent manuscript, well written, whose conclusions are based on a thorough study of the underlying mechanisms of the members of the ciliary tip complex on MT dynamics. The quality of the work is very high, and the results are very important for our understanding of this very specific slow MT growth observed not only in cilia but also on their associated structure, the centriole.

Overall, this manuscript provides a very nice dissection of the tip module's mechanism of action, with very well-described interactions. The work is clean, with all controls present, beautiful and very well done with appropriate statistical analyses and will be useful to a wide target audience such as the one of NSMB, especially for those working on MTs, cilia and centrioles.

We thank the reviewer for their supportive comments. We have addressed the specific questions below.

I only have a small number of minor comments that need to be addressed before publication.

1-Page 5: the authors mention that TOGARAM1 decreases MT growth rate and that this is unexpected but do not comment more on this? Could they try to expand a bit on this?

We have no definitive explanation for this observation, which is indeed unexpected, because structurally, TOGARAM1 resembles the well-known microtubule polymerase XMAP215/chTOG, which strongly accelerates tubulin addition, but can also catalyze tubulin removal and promote catastrophes (Brouhard et al, 2008, doi: 10.1016/j.cell.2007.11.043; Farmer et al, 2021, doi: 10.1083/jcb.202012144). Since TOGARAM1 can convert blocked microtubule tips to slowly growing ones, it does have properties of a microtubule polymerase. However, the fact that it inhibits, rather than stimulates tubulin addition in the absence of pausing factors, makes it more similar to CLASPs, TOG-domain-containing proteins which also mildly inhibit microtubule growth but at the same time potently prevent depolymerization and promote re-growth of microtubule ends that start losing their GTP cap (Aher et al, 2018, doi: 10.1016/j.devcel.2018.05.032). We included this point in the Discussion on page 13:

“TOGARAM1 can thus be regarded as a slow MT polymerase, which shows some functional similarities to CLASPs; TOG domain proteins that mildly reduce growth rate but potently promote re-growth of microtubule ends that start losing their GTP cap⁴⁸.”

2- the data is very convincing. For the graphs Figure 1d, e, f, the labels are only in f (CCDC66, etc) which makes it a bit difficult to read. I know that the space is limiting but may be boxing the lines of the column bars with a dedicated color for the protein may help (might also be too many colors to be tested, may be use patterns instead of colors for the pause to shrinkage, etc). could the authors try to make the reading on the graph easier if possible?

Additional x-axis labels have been added below figure 1d, dashed lines between assays with and without EB3, and pattern bars for all conditions with EB3. We hope this makes the quantification in figures 1d-f easier to read.

Figures 1d-f. Parameters of MT plus-end dynamics in the presence of 15 μM tubulin alone or with 20 nM EB3 in combination with indicated concentrations of proteins (from kymographs shown in **b**, **c**, **i-p**, and **S1d**).

3-FRAP of CEP104 shows no recovery (Fig 2c)- CEP104 does not exchange at blocked MTs plus end. What about CEP104 FRAP experiment in the context of the full module (the five proteins together as in Fig 6/7)? Would you see some recovery since the authors propose that the tip complex form a flexible interaction network? Or do you think that CEP104 will still not exchange under these conditions and stays at the MT tip?

We thank the reviewer for suggesting this insightful experiment, which shows that recovery (and thus exchange) of CEP104 is indeed observed in the context of the whole module. The following data has been included in Fig S6f and accompanying text in the results on pages 10-11.

New figure S6f. FRAP analysis of CEP104 at slow growing MT plus ends with the entire ciliary tip module. Arrowhead marks point of photobleaching in representative kymograph (top), scale bars 2 μ m and 60 s. Plot (bottom) shows average curve with exponential fit. Number of FRAP measurements, n=19 . Error bars represent s.e.m..

Results text, page 10:

“Interestingly, unlike in the conditions where CEP104 blocked MT plus ends (Fig. 2c), FRAP of CEP104 in these conditions showed partial, albeit very slow, recovery over the course of 10 minutes, indicating in these conditions some turnover of CEP104 is needed for tubulin addition (Fig. S6f).”

4-This is more a discussion point: The authors convincingly show how the tip complex can modulate slow axonemal MT growth. They mention that some of these proteins are at centrioles. Do they think such a complex (like the CP110/CPAP one which also display a “plug” configuration) would be at act also during centriole assembly ensuring a very slow growth of the procentriole? Which members of the complex do not localize at centrioles? Can the authors extend the discussion of this point further to what is already written. Do they

think that CEP104 will also be in the CP110 complex at centriole since the two proteins can interact? Or do they think that specific complexes exist to perform similar function at centrioles and at the plus end of axoneme?

This is an excellent point. CEP104 is also present at the distal ends of centrioles and interacts with the centriolar cap proteins CP110 and CEP97 (Jiang *et al.*, 2012, doi: 10.1016/j.cub.2012.07.047). Furthermore, CEP104 was recently shown to regulate centriolar growth in *Drosophila* (Ryniawec *et al.*, 2023, doi:10.1016/j.cub.2023.08.075). This argues for some mechanistic overlap between the regulation of ciliary and centriolar microtubule tips. On the other hand, it is established that CP110 needs to be removed from the distal end of the centriole for cilia to form (Spektor *et al.*, 2007, doi: 10.1016/j.cell.2007.06.027; Goetz, *et al.*, 2012, doi: 10.1016/j.cell.2012.10.010; Huang *et al.*, 2018, doi: 10.1038/s41467-018-06990-9; Prosser and Morrison, 2015, doi: 10.1083/jcb.201411070.), which points to clear differences between the formation of centriole and ciliary microtubules. We have included a note about this in the Discussion on page 16:

Accompanying text, Discussion page 16:

“MT stabilization from the luminal side combined with the reduction of protofilament curvature and occlusion of the longitudinal interface of β -tubulin might thus be a common mechanism for generation of very stable and slowly growing MTs, such as those of cilia and centrioles. Interestingly, although growth of centrioles and cilia is mostly controlled by distinct sets of proteins, CEP104 can bind to CP110 at the distal tip of the centriole^{16, 64, 65} and plays a role in centriole elongation in flies²⁴.”

Point-by-point answers to the Reviewers' Comments:

Reviewer #2 (Remarks to the Author):

The authors addressed all of my concerns. It's good to put down the statistics of cryoET in so the audience know what were analyzed.

This in vitro work is the first step towards understanding the functions of so many ciliary MT-binding proteins. While I agree with the other reviewer that in vitro may not recapitulate everything that happens in cells, I still think this in vitro work represents the state-of-the-art understanding and provides new hypotheses to be tested. It also fills a long-lasting gap in the field of ciliary biochemistry.

We thank the reviewer for the positive comments.

Reviewer #3 (Remarks to the Author):

In this revision, the authors addressed the concerns and suggestions of reviewers (me and two other reviewers) with more experiments and analysis, such as quantifying the cork density in the cryo-ET and stoichiometry variation in the tip proteins for the microtubule dynamic assays. They also added texts in the results and discussions to clarify different points.

While there are still things unclear about the cryo-ET data, such as the microtubule service is very clean without many densities binding outside, the cryo-ET is only a part of the manuscript. Therefore, I believe the author addressed the concern appropriately in this revision. The paper is good for publication for their findings.

We thank the reviewer for the positive comments.

A small suggestion. In methods, the authors need to show how many tomograms were collected and if different datasets are collected (biological replicate or technical replicate). This will support the reproducibility as well.

This information has been included in the Methods section on page 35. Protofilament tracing and analysis was performed on two biological replicate datasets for the control condition (dataset 1, n=8 MTs; dataset 2, n=18 MTs) and three biological replicate datasets for the CTM condition (dataset 1, n=20 MTs; dataset 2, n=6 MTs; dataset 3, n=7 MTs).

Furthermore, if possible, the cryo-ET data or at least representative data should be deposited or shared on EMPIAR or EMDB.

Cryo-electron tomography data generated for the paper has been deposited to EMDB and EMPIAR. All tomographic subvolumes on which MT protofilament tracing was performed, as well as representative full volume tomograms, are available at EMDB through the following accession codes: EMD-52359 for the control condition (microtubules grown with EB3 alone) and EMD-52360 for the CTM condition (microtubules grown with the CTM and EB3). The raw tilt series data

and the protofilament tracing models are available at EMPIAR through the following accession codes: EMPIAR-12498 for the CTM condition (microtubules grown with the CTM and EB3) and EMPIAR-12499 for the control condition (microtubules grown with EB3 alone).

Reviewer #4 (Remarks to the Author):

The authors responded satisfactorily to the points raised during the review process. The new provided results are convincing. I am convinced by the revised version of the manuscript and recommend it for publication in NSMB.

We thank the reviewer for the positive comments.